# Exploring the influence of citizen involvement on the assimilation of crowdsourced observations: a modelling study based on the 2013 flood event in the Bacchiglione catchment (Italy)

5    Maurizio Mazzoleni[1], Vivian Juliette Cortes Arevalo[2], Uta Wehn[1], Leonardo Alfonso[1], Daniele Norbiato[3], Martina Monego[3], Michele Ferri[3], Dimitri P. Solomatine[1,4,5]

[1]Integrated Water Systems and Governance Department, IHE Delft Institute for Water Education, Delft, 2611AX, the Netherlands
[2]Water Engineering and Management, University of Twente, Enschede, 7522 NB, the Netherlands
10    [3]Alto Adriatico Water Authority, Venice, Italy
[4]Water Resources Management department, Water Problems Institute, Russian Academy of Sciences, Moscow, Russia
[5]Water Resources Section, Delft University of Technology, Delft, 2628 CD, the Netherlands

*Correspondence to*: M. Mazzoleni (m.mazzoleni@un-ihe.org)

**Abstract**

15    To improve hydrological predictions, real time measurements derived by traditional physical sensors are integrated within mathematic models. Recently, traditional sensors are being complemented with crowdsourced data (social sensors). Despite measurements from social sensors can be low-cost and more spatially distributed other factors like spatial variability of citizen involvement, decreasing involvement over time, variable observations accuracy and feasibility for model assimilation play an important role for accurate flood predictions. Just few studies have investigated the benefit of assimilating uncertain 20    crowdsourced data in hydrological and hydraulic model. In this study, we investigate the usefulness of assimilating crowdsourced observations from a heterogeneous network of static-physical, static-social and dynamic-social sensors. We assess improvements in the model prediction performances for different spatial-temporal scenarios of citizens' involvement levels. To that end, we simulate an extreme flood event occurred in the Bacchiglione catchment (Italy) in May 2013 using a semi-distributed hydrological model with the station at Ponte degli Angeli (Vicenza) as prediction/validation point. A 25    conceptual hydrological model is implemented by the Alto Adriatico Water Authority and it is used to estimate runoff from the different sub-catchments, while a hydraulic model is implemented to propagate the flow along the river reach. In both models, a Kalman filter is implemented to assimilate the crowdsourced observations. Synthetic crowdsourced observations are generated for either static social or dynamic social sensors because these measures were not available at the time of this study. We consider two sets of experiments: i) assuming random probability of receiving crowdsourced observations and ii) using 30    theoretical scenarios of citizen motivations, and consequent involvement levels, based on population distribution. The results demonstrate the usefulness of integrating crowdsourced observations. First, the assimilation of crowdsourced observations

located at upstream points of the Bacchiglione catchment ensure high model performance for high lead time values, whereas observations at the outlet of the catchments provide good results for short lead times. Second, biased and inaccurate crowdsourced observations can significantly affect model results. Third, the theoretical scenario of citizens motivated by their feeling of belonging to a "community of friends" has the best effect in the model performance. However, flood prediction only improved when such small communities are located in the upstream portion of the Bacchiglione catchment. Finally, decreasing involvement over time leads to a reduction of the model performance and consequently inaccurate flood forecasts.

## 1 Introduction

A challenge for water management is the reduction of risk related to extreme events such as floods. Flood management needs timely provision of early warning information, for example, to operate control structures and to regulate water levels. Reliable and accurate streamflow simulation and water level prediction by means of hydrological and hydraulic models are therefore of uttermost importance. However, model performance and related predictions are inherently uncertain due to the lack of reliable and sufficient observational data, lack of understanding of the natural hydrological and hydraulic processes, and limitations and assumptions of the modelling system (Merz et al., 2010, p 514).

Different attempts have been made to improve the accuracy of flood model predictions for operational early warning. In particular, data assimilation techniques have been extensively used (Liu et al., 2012). Data assimilation is a common method for updating model input, parameters, states or outputs. It is used to integrate real-time observations of hydrological variables (WMO, 1992; Refsgaard, 1997) while accounting for the uncertainties in both model and observed data (McLaughlin, 1995; Robinson et al., 1998; McLaughlin, 2002; Madsen and Skotner, 2005; Lahoz et al., 2010; Liu et al., 2012). In operational early warning systems, only observed data derived by static-physical (StPh) sensors are used, as described in Liu et al. (2012). However, recent studies have demonstrated that water system models could improve their performances with the assimilation of observations from multiple sources such as in-situ and remote sensors, and other hydrologic variables such as soil moisture and streamflow (Aubert et al., 2003; McCabe et al., 2008; Pan et al., 2008; Lee et al., 2011; Montzka et al., 2012; Pipunic et al., 2013; Andreadis et al., 2015; Lopez Lopez et al., 2015; Rasmussen et al., 2015). Those studies have also shown that data assimilation applications require specific, frequent and high quality measurements.

In parallel, the availability of recent technological advances to the public has strengthen the idea of involving people in data collection. This idea is not limited to the data collection of flood or real-time information and various terms have been used in scientific literature (Wehn and Evers, 2015). In Natural Science this idea is known as 'citizen science' (Silvertown, 2009); in Geography, 'volunteer geographic information, VGI' (Goodchild, 2007) and 'crowdsourcing geospatial data' (Heipke, 2010), and in Computer Science 'people-centric sensing' (Campbell et al., 2006) and 'participatory sensing' (Höller et al., 2014). Other terms explicitly emphasise the involvement of the public, for instance the 'value of information and public participation' (Alfonso, 2010), 'public computing' (Anderson, 2003) and 'community data collection' (Aanensen et al., 2000).

Crowdsourcing particularly refers to the involvement of a large, often undefined and diverse group of people in data collection and/or data analysis and can be mediated via information technologies and online tools or platforms (Xintong, Hongzhi, Song, & Hong, 2014). In this study, we refer to crowdsourced (CS) citizen based-observations to as the involvement of citizens in general (either experts or not) in collecting water level observations at a particular location via a smartphone app upon request of the water authorities.

Some previous studies have attempted to use CS citizens-based observations in water system models since a more spatially distributed coverage can be achieved. (Alfonso 2010; Fava et al. 2014; Smith et al. 2015; Fohringer et al., 2015; Gaitan et al., 2016; Giuliani et al., 2016; de Vos et al., 2017; Rosser et al., 2017; Schneider et al., 2017; Starkey et al., 2017; Yu et al., 2017). In Fava et al. (2014) a methodology for flood forecasting integrating VGI and wireless sensor networks is proposed. Smith et al. (2015) and Fohringer et al. (2015) proposed frameworks for real-time flood monitoring using information retrieved from social media. In both studies, the observation filtering process was one of the main challenges. Rosser et al. (2017) proposed a data fusion method to rapidly estimate flood inundation extent using observations from remote sensing, social media and high resolution terrain mapping. Yu et al. (2017) validated the results of an urban hydro-inundation model (surface water related flooding) with a crowdsourced dataset of flood incidents. In a similar fashion, Starkey et al. (2017) demonstrated the value of community-based observations for modelling and understanding the catchment response. In particular, they have showed the significant improvement in the spatial and temporal characterisation of the catchment response by integrating local network of community-based observations together with traditional network rather than using traditional observations only. Recently, Herman Assumpção et al. (2017) provided a detailed review of the studies in which citizen observations are used for flood modelling applications.

However, none of the previous studies assessed the usefulness of CS observations in improving flood predictions, nor taken into account the variable distribution, intermittency and, potentially, lower-quality of citizen-based data (Shanley et al., 2013; Buytaert et al., 2014; Lahoz and Schneider, 2017). First attempts are reported in Mazzoleni et al. (2015; 2017a and 2017b) and Mazzoleni (2017). In those studies, the authors investigated the effects on flood prediction in assimilating real-time (synthetic) CS observations in hydrological models. However, in the former studies the authors did not investigate the effects of assimilating (synthetic) CS observations in hydraulic models. Furthermore, the authors did not consider (theoretical) scenarios of citizen involvement, nor the simultaneous assimilation of CS observations from static and dynamic social sensors. For this reason, the main objective of this study is to assess the usefulness of assimilating CS observations in model-based predictions of flood events. We analyse a flood event which occurred in May 2013 in the Bacchiglione catchment (Italy). Static-physical (StPh), static-social (StSc) and dynamic-social (DySc) sensors are considered in this study. Synthetic CS observations of water level are assimilated in a cascade of hydrological and hydraulic models since real CS measurement are not yet available for this particular study site. Two sets of experiments of theoretical scenarios are analysed. Citizen involvement level (CIL) is further defined as the probability of receiving a CS observation based on the citizen's own interest or intention in collecting water levels. We assume that CIL mainly limit the intermittency or timely availability of observations. The achievement of the

paper's objective is a step forward in understanding the effect of public involvement on the possible improvement of hydrological and hydraulic models, with methods that can be replicated in other fields.

## 2 Case study

### 2.1 The Bacchiglione catchment

The Bacchiglione catchment (North East Italy, see Figure 1) is one of the case studies in which WeSenseIt (http://wesenseit.eu,/ WSI) Citizen Observatory of Water Project developed and tested innovative static and low-cost mobile sensors (Ciravegna et al., 2013). The main goal of the WSI project was to allow active citizens to support the work of water authorities by providing CS observations. Innovative static sensors were strategically integrated into the existing monitoring networks for collecting physical and CS data. Low-cost mobile sensors were developed such as a mobile phone app, which uses a Quick Response
(QR) code for geographical referencing and allows to send among others, flood reports and water level ($W_L$) observations. In addition, WSI project set up a pilot platform in which CS observations collected with this app can be sent. However, this pilot is not yet operational and CS observations are not yet available (see details of the testing of this pilot in Section 2.3). In this research, only $W_L$ data is assimilated.

This research focuses on the upper part of the Bacchiglione catchment which flows into the Adriatic Sea at the South of the
Venetian Lagoon. The case study has an overall extent of about 450km$^2$ with a river length of approximately 50km. The three main tributaries are the Timonchio River on the East side and Leogra and Orolo Rivers on the West side. The main urban areas are located close to the outlet section of the case study area, the city of Vicenza. The Alto Adriatico Water Authority (AAWA) is currently using an operational semi-distributed hydrological and hydraulic model for early warning (Ferri et al., 2012, Mazzoleni et al., 2017a). Forecasted and measured precipitation time series are available for a flood event that occurred in
May 2013. The forecasted precipitation time series are provided by the Cosmo-LAMI model, a regional model that provides numerical prediction over the national territory at 7 km resolution and three-day time interval. Currently, AAWA is performing quality control on the forecasted data before using them in the Bacchiglione flood early warning system. The measured precipitations are supplied and validated by Veneto Regional Agency of Environmental Prevention and Protection (ARPAV). The event of May 2013 is considered to be significant due to its high intensity, which resulted in several traffic disruptions at
various locations upstream Vicenza. In this study, we assess the usefulness of assimilating CS $W_L$ (synthetic) observations in the hydrological and hydraulic models to improve the model performance and consequently flood prediction.

### 2.2 Sensors classification

Despite that CS observations were not operational nor available in the case study, we analysed the characteristics of each sensor to generate the synthetic $W_L$ observations that we assimilated for the flood event of 2013. We considered three types of
sensors to measure $W_L$, static physical (StPh), static social (StSc) and dynamic social sensors (DySc) sensors. Currently, only

StPh sensors are used by AAWA to provide daily flood forecast in the Bacchiglione catchment. This paper sections aims at describing the characteristics of these sensors in terms of spatial coverage and accuracies.

The StPh sensors are traditional physical sensors such as water level ultrasonic sensors. StPh have a fixed location, and a regular measurement interval. Data from StPh sensors are validated by ARPAV. Observational error depends on how well documented is the cross section where the StPh sensor is located, random and bias errors due to sensor characteristics. Despite of the potential observational error, we assume high accuracy level as the observation is automatically generated by the sensor therefore not affected by the variability of CS data.

StSc have a higher spatial distribution along the river reach but are characterized by intermittent CS observations. The StSc sensors are staff gauges at a safe, strategic and accessible locations along the river reaches. Citizens can report observations using these static sensors to estimate $W_L$ values. According to the data collection tool, CS observations can come in a variety of formats either quantitative or qualitative, which is often one of the biggest challenges when involving citizens. Automatic mechanisms for data processing can be implemented. For example, whenever photos are collected can be automatically analysed using image recognition methods as proposed by van Overloop and Vierstra (2015) and Le Boursicaud et al. (2015). In this case a reference gauge must be available. The WSI mobile phone app will be used to send quantitative measurements (water level) observed at a specific staff gauge. Photos and videos are not supported by the WSI app. The geographical referencing will be provided by means of QR codes together with associated date/time. The WSI mobile app is equipped with a filter that automatically discards the water level measurements that fall outside the associated range to the staff gauge.

DySc sensors are characterized by a not fixed locations. Water level observations at a particular location via a smartphone application can be requested/discouraged by water authorities according to the accessibility of the location. A possible method for measuring flow using DySc sensors is described in Lüthi et al. (2014). The authors proposed an approach based on particle image velocimetry to estimate, with acceptable accuracy water level, surface velocity and runoff in open channels. However, this approach requires a priori knowledge of the channel geometry at the location of the measurement, which is one of the main sources of uncertainty. For this reason, in this paper it is assumed that DySc sensors have lower accuracy than StSc sensors. Another example of DySc sensors is reported in Michelsen et al. (2016) where water level time series are derived from the analysis of YouTube videos. It is worth noting that the WSI mobile app does not allow for automatic retrieve of flow information from photos and video as proposed in Lüthi et al. (2014).

As reported in Table 1, $W_L$ observations have different characteristics of temporal availability and accuracy based on the adopted sensor and changes in the cross section. Regardless of the type of social sensor either experts or amateur, we acknowledge that the data accuracy and intermittency of CS observations can be affected by various factors. Source of errors in observations include but are not limited to (Cortes Arevalo, 2016; Kerle & Hoffman, 2013; Le Coz et al., 2016): i) the expertise level (training and experience is still required to read a gauge, take a picture and use the mobile applications developed), ii) type and format of CS observation based on sensor classification and data collection procedure ($W_L$ measurement and photo with reference to a staff gauge vs a photo with reference to a neighbouring object) iii) the specific conditions at the reporting location (accessibility, visibility and environmental conditions). Intermittency (temporal

availability) of the CS observations is directly related to CIL, i.e. the probability of receiving a CS observation. In addition, CS observations imply the filtering and integration of a variety of formats and information types, which requires to develop suitable tools for data collection and processing (Kosmala, Wiggins, Swanson, & Simmons, 2016).

5 **Table 1.** General characteristics of type of observations based on sensor classification

| Sensor type | Type of observation | Location | Time of availability | Observational error | Example reference | Assumed accuracy level |
|---|---|---|---|---|---|---|
| **Static Physical (StPh)** | Water level time series | Fixed, generally in key inlet or outlets | Each model time step | - Missing data due to for example unexpected damage or lack of maintenance<br>- Observational noise due to flow conditions and water level below or above the optimum range.<br>- Missing or not representative rating curve due to changes in the cross section. | Irrigation Training and Research Center, 1998, p. 58 | High |
| **Static Social (StSc)** | Water level and photo of the river gauge. | Fixed but distributed at strategic points along the river reach | Intermittent, according to CIL | - Same as StPh<br>- Inaccurate reading of the river gauge<br>- Inaccurate photo limiting validation<br>- Unknown expertise level of the citizen reporting | Le Boursicaud, Pénard, Hauet, Thollet, & Le Coz, 2016, pp. 95–99; Le Coz et al., 2016, p. 770 | Medium |
| **Dynamic Social (DySc)** | Photo and water level estimation by means of mobile app | Variable | Intermittent, according to CIL and accessibility level to the river reach | - Same as StPh<br>- Same as StSc but inaccurate estimation of the flow using mobile app<br>- Unknown (irregular) cross section and river bank conditions at the reported location | | Low |

## 2.3 Citizen involvement in the Bacchiglione catchment

Gharesifard, et al. (2017) categorized participants into netizens, citizen scientists and volunteers to accordingly distinguish: i) unawareness about their implicit involvement and contribution to monitoring networks (netizens); ii) explicit and intentional involvement in data provision (citizen scientists) and iii) the involvement of individuals or groups that are systematically targeted and recruited to participate in data provision with pre-defined goal(s) (volunteers).

In the framework of the WeSenseIt project, an exercise was carried out with volunteers who were providing water level observations via the smartphone app, from a limited number of locations to test the pilot set up. However, due to the limited number of participants, duration and testing goal of the exercise, no formal assessment of citizen involvement could be undertaken. For this reason, we propose theoretical involvement scenarios to represent the hypothetical situations whenever citizens are fully or partially involved in the Bacchiglione catchment. In the numerical simulations performed in this study, we did not make distinction between citizen expertise (expert or amateur) and involvement type (citizen scientists or volunteers). We do not refer to the engagement process (how to get citizens involved) but rather to the involvement level (probability of receiving a CS observation based on the citizen's own interest or intention in collecting water levels). In fact, motivations and involvement levels are the only variables that differentiate the citizens, as described in the next sections

## 3 Modelling tools

### 3.1 Semi-distributed hydrological model

In order to implement the semi-distributed model, the Bacchiglione catchment is divided into different sub-catchments and the so-called inter-catchments which streamflow contributions run into the main river channel up to the urbanized area of Vicenza. In the schematization of the Bacchiglione catchment (see Figure 1), the location of the StPh and StSc sensors corresponds to the outlet section of the three main sub-basins, Timonchio, Leogra and Orolo. The remaining sub-basins are considered as inter-catchments. The rainfall-runoff processes within each sub-catchment and inter-catchment are represented by the conceptual hydrological model developed by AAWA. In the case of the main river channel, a hydraulic model is used to propagate the flow down to the gauge station of PA in Vicenza. The river reach is divided into several reaches according to the location of the internal boundary conditions. We use hydrological outputs as upstream (from sub-catchments) and internal boundary conditions (from inter-catchments). Figure 1 shows that the output of the hydrological model (red arrows) are boundary conditions for the proposed hydraulic model.

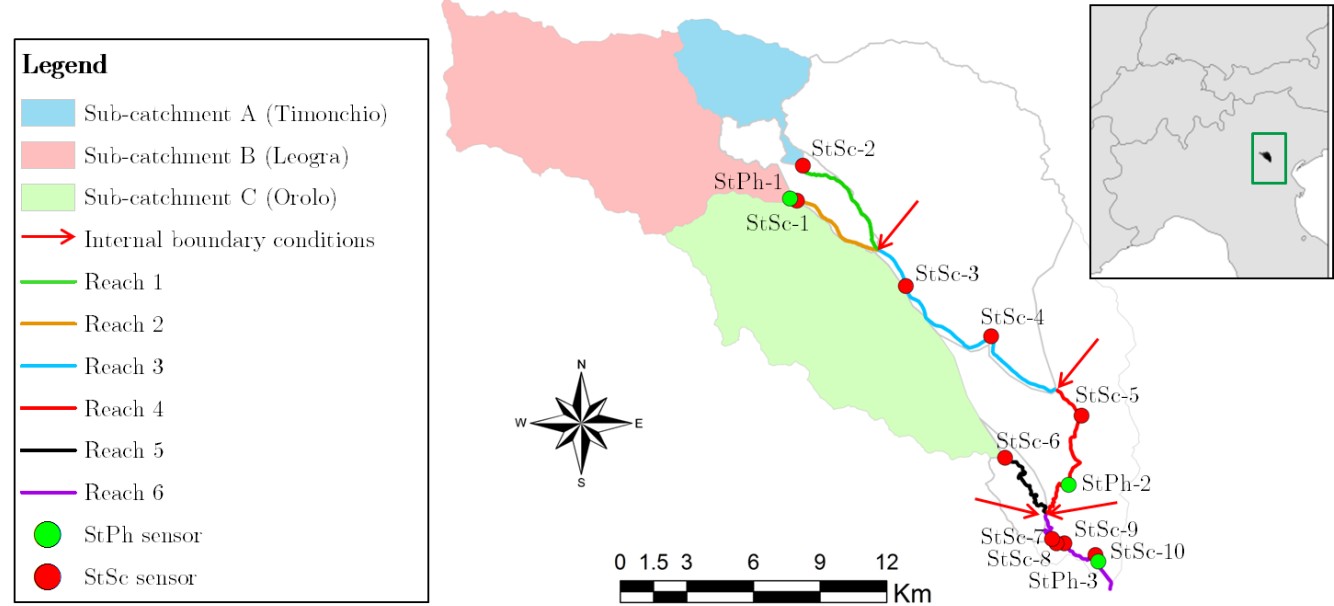

**Figure 1.** Spatial distribution of the sub--catchments, river reaches, and StPh and StSc sensors implemented in the catchment by AAWA. The prediction point of Ponte degli Angeli (PA) corresponds to the StPh-3 sensor.

### 3.1.1 Hydrological modelling

5 The hydrological model used in this study is a part of the early warning system implemented and used by AAWA. We briefly relate to the model equation here as a detailed description is available in Ferri et al. (2012) and Mazzoleni et al. (2017a). Precipitation time series is the only input. The water balance is applied to a generic control volume of active soil, at the sub-basin scale, to mathematically represent the processes related to runoff generation processes such as surface, sub-surface and deep flow.

$$S_{W,t+dt} = S_{W,t} + P_t - R_{sur,t} - R_{sub,t} - L_t - E_{T,t} \qquad (1)$$

where $S_{W,t}$ is the water content at time $t$, $P$ is the precipitation component, $E_T$ the evapotranspiration, $R_{sur}$, the surface runoff, $R_{sub}$ the subsurface runoff and $L$ is the deep percolation. Temperature is used for the estimation of the real evapotranspiration, which is calculated using the formulation of Hargreaves and Samani (1985). The routed contributes of the surface flow $Q_{sur}$, sub-surface flow $Q_{sub}$ and deep flow $Q_g$ are derived from $R_{sur}$, $R_{sub}$ and $L$ by means of the conceptual framework of the linear
15 reservoir model.

Calibration of the hydrological model parameters, including the parameters of the linear reservoir model for $Q_{sub}$ and $Q_g$, is performed by AAWA minimizing the error between the observed and simulated $W_L$ values at Ponte degli Angeli (PA) for a period between 2000 and 2010 (Ferri et al., 2012). In order to apply the data assimilation approach and properly integrate crowdsourced $W_L$ observations within the mathematical model, it is necessary to represent the previous dynamic system in a
20 state-space form, i.e.:

$$\mathbf{x}_t = M(\mathbf{x}_{t-1}, \vartheta, \boldsymbol{I}_t) + w_t \tag{2}$$

$$\mathbf{z}_t = H(\mathbf{x}_t, \vartheta) + v_t \tag{3}$$

where, $\mathbf{x}_t$ and $\mathbf{x}_{t-1}$ are the model state vectors respectively at time $t$ and $t$-1; $M$ is the model operator, $\boldsymbol{I}_t$ is the vector of the model inputs; $H$ is the operator which maps the model states into the model output $\mathbf{z}_t$. The terms $w_t$ and $v_t$ indicate respectively the

system and measurements errors which are assumed normally distributed with zero mean and covariance $S$ and $R$. In case of the hydrological model used in this study, the states are identified in $x_S$, $x_{sur}$, $x_{sub}$ and $x_L$, i.e. the states to $S_W$ and to the linear reservoir generating $Q_{sur}$, $Q_{sub}$ and $Q_g$. In Mazzoleni et al. (2017a), sensitivity analysis is carried out by perturbing the model states ±20% around the true state every time step in order to find out to which model states the output is more sensitive. The study shows that model output is most sensitive to $x_{sur}$. For this reason, we decide to update only the model state $x_{sur}$, which is

related to the linear reservoir, so the state-space form can be expressed as follows:

$$\mathbf{x}_t = \boldsymbol{\Phi}\mathbf{x}_{t-1} + \boldsymbol{\Gamma}I_t + w_t \tag{4}$$

$$\mathbf{z}_t = \mathbf{H}\mathbf{x}_t + v_t \tag{5}$$

where $\mathbf{x}$ is the vector of the model states (stored water volume in m$^3$), $\boldsymbol{\Phi}$ is the state-transition matrix, $\boldsymbol{\Gamma}$ is the input-transition matrix, $\mathbf{H}$ is the output matrix. In this case, the model output $\mathbf{z}$ is expressed as streamflow $Q$ at the outlet section of the sub-

catchment or inter-catchment. The detailed description of data assimilation in linear systems and the ways the matrices $\boldsymbol{\Phi}$, $\boldsymbol{\Gamma}$ and $\mathbf{H}$ are built can be found, e.g., in Szilagyi and Szollosi-Nagi (2010).

### 3.1.2. Hydraulic modelling

Flood propagation along the main river channel is represented using a Muskingum-Cunge (MC) model (Cunge, 1969; Ponce and Chaganti, 1994; Ponce and Lugo, 2001; Todini, 2007); it is based on the mass balance equation applied over a prismatic

section delimited by the upstream and downstream river section. As described in Cunge (1969) and Todini (2007), a four point time centered scheme can be applied to numerically solve the kinematic routing equation, and to derive a first order approximation of a kinematic wave model and express the MC model as:

$$Q_{t+1}^{j+1} = C_1 Q_t^j + C_2 Q_t^{j+1} + C_3 Q_{t+1}^j \tag{6}$$

where $t$ and $j$ are the temporal and spatial discretization and $Q$ is the streamflow; $C_1$, $C_2$ and $C_3$ are the routing coefficients,

which are function of the geometry of the cross-sections and wave celerity, calculated at each time step $t$ following the approach proposed by Todini (2007) and reported in detail by Mazzoleni (2017). It is worth noting that in this formulation of MC model, the only model parameter is the Manning coefficient of the river channel considered in the estimation of the wave celerity. In addition, MC model is implemented, independently, along each one of the six river reaches represented in Figure 1.

As in the case of a hydrological model, to apply the data assimilation method, the state-space form of the hydraulic model is

used as well. The state and observation process equations are similar to the ones described in Eq.(4) and (5). In case of the

hydraulic model, the model state vector is defined as $\mathbf{x}_t=(Q_t^1, Q_t^2,..Q_t^j,...,Q_t^N)$, where $Q$ is the discharge along the river in m³/s, while the input matrix is $\mathbf{I}_t=(Q_t^1, Q_{t+1}^1)$ being $Q^1$ the discharge at the upstream boundary condition. The state-transition $\mathbf{\Phi}$ and input-transition $\mathbf{\Gamma}$ matrixes are calculated following the approach derived by Georgakakos et al. (1990). In the observation process of the hydraulic model, $z$ represents the flow along the river channel, while $\mathbf{H}$ is output matrix equal to $[0\ 0\ \dots\ 1]^T$ in

case of flow measurements at the outlet section of the river reach. In this study, due to the varying position of social sensors, the matrix $\mathbf{H}$ changes accordingly at each time step. The Manning equation is used to estimate the $W_L$ in the river channel knowing the value of flow at each spatial discretization step, considered 1000m in order to guarantee the numerical stability of the MC model scheme.

## 3.2 Data assimilation

The Kalman Filter (KF, Kalman 1960) is a mathematical tool widely used to integrate real-time noisy observations, in an efficient computational (recursive) algorithm, within a dynamic linear system resulting in the best state estimate with minimum variance of the model error. In Liu et al. (2012), a detailed review of KF and other type of data assimilation approaches is reported. The first step in the KF procedure is the forecast of the model state vector, following Eq.(4), and the covariance matrix is expressed as:

$$\mathbf{P}_t^- = \mathbf{\Phi}\mathbf{P}_{t-1}^+\mathbf{\Phi}^T + \mathbf{S}_t \tag{7}$$

where the superscript − indicates the forecasted model error covariance matrix $\mathbf{P}$ and the superscript + indicates the updated state value coming from the previous time step. When an observation $z^o$ becomes available, the second (update) step of the KF is executed, in which the forecasted model states $\mathbf{x}$ and covariance $\mathbf{P}$ are updated as:

$$\mathbf{x}_t^+ = \mathbf{x}_t^- + \mathbf{K}_t(z_t^o - \mathbf{H}_t z_t^o) \tag{8}$$

$$\mathbf{P}_t^+ = (\mathbf{I} - \mathbf{K}_t\mathbf{H}_t)\mathbf{P}_t^- \tag{9}$$

$$\mathbf{K}_t = \mathbf{P}_t^-\mathbf{H}_t^T(\mathbf{H}_t\mathbf{P}_t^-\mathbf{H}_t^T + \mathbf{R}_t)^{-1} \tag{10}$$

where $\mathbf{K}$ is the Kalman gain matrix (the higher its values, the more confidence KF gives to the observation $z^o$ and vice versa). Due to the fact that along the river channel only $W_L$ observations are provided, the manning equation is used to express the vector $z^0$ as streamflow based on the river cross-section geometry.

In this study, CS observations are considered. As already mentioned, such observations can be irregular both in time and in space. In order to consider the intermittent nature in time within the KF, the approach proposed by Cipra and Romera (1997) and Mazzoleni et al. (2015) is adopted. According to this approach, when no observation is available, the model state vector $\mathbf{x}$ is estimated using Eq.(4), while the model error covariance $\mathbf{P}$ is left unchanged:

$$\mathbf{P}_t^+ = \mathbf{P}_t^- \tag{11}$$

It is worth noting that in case of a hydraulic model, the state variables at each reach are updated independently

### 3.3 Synthetic observations

In operational practice, $W_L$ values are converted into streamflow values to be then assimilated within hydrological models. This is usually done using the available rating curves at the sub-catchment outlets. On the other hand, $W_L$ data usually can be directly assimilated in hydraulic models, but the problem is that the MC model used in this study requires flow information rather than $W_L$. For this reason, the synthetic $W_L$ observation at a certain random location (DySc sensor) is converted into streamflow by means of the Manning equation if no rating curve information is available. In fact, it is quite unlikely to have the information of the rating curve at a random location of the CS observation provided by DySc sensors in real world applications. When there are no data regarding the cross-section, assumptions should be made about a rectangular cross-section with a given width and depth. However, this approach will introduce significant uncertainty in river flow estimation. A possible solution is the use of mobile apps able to automatically retrieve of flow information from photos and video as proposed in Lüthi et al. (2014), Overloop and Vierstra (2015) and Le Boursicaud et al. (2015). We believe that this type of mobile apps will increasingly become available (at reasonable low costs) to citizen in order to easily measure river flow.

Due to the lack of distributed CS observations at the time the considered flood event occurred, synthetic $W_L$ observations are used (Mazzoleni et al., 2017a). In order to generate such synthetic observations, the observed time series of precipitation during the considered flood event are used as input for the hydrological models of the sub-catchments and inter-catchments to generate synthetic discharges and then propagate them with the hydraulic model down to the prediction point of PA (corresponding to the sensor StPh-3 in Figure 1). In this way, the synthetic $W_L$ values at the outlet of the sub-catchments/inter-catchments and at each spatial discretization of the six reaches of the Bacchiglione River are estimated, and assumed as observed variables in the assimilation process. In meteorology, this kind of approach is often called "observing system simulation experiment" (OSSE), as described for example by Arnold and Dey (1986), Errico et al. (2013) and Errico and Privé (2014).

Regarding the observation error, as described in Weerts and El Serafy (2006), Rakovec et al. (2012), and Mazzoleni (2017), the covariance matrix $R$ is assumed to be:

$$R_t = \left( \alpha_t \cdot Q_t^{synth} \right)^2 \tag{12}$$

where $\alpha$ is a variable related to the accuracy level of the measurement. The accuracy (i.e. degree to which the measurement is correct overall) is subjected to random error and bias or systematic errors (Bird et al., 2014). Moreover, for $W_L$ observations accuracy levels vary temporally, spatially and for each physical or social sensor. Table 2 summarises the distribution of the coefficient $\alpha$ of the observational error of Eq.(12). The distribution of the coefficient $\alpha$ does not pretend to be exhaustive in accounting the different accuracies between observations coming from physical and social sensors but a first and simplified approximation that is aspect for further research (see details in section 2.2 and Table 1).

Although there are many sources of uncertainty in the indirect estimation of streamflow, in case StPh sensors it is assumed that the rating curve estimation is the main source of uncertainty to properly estimate the streamflow given a certain $W_L$ value.

In fact, for the StPh sensors used in this study the instrument precision is about 0.01 m. As described in Weerts and El Serafy (2006) and Rakovec et al. (2012), the coefficient $\alpha$ is assumed equal to 0.1, constantly in time and space.

On the other hand, due to the unpredictable accuracy of the CS observations coming from the sensors StSc and DySc sensors, the coefficient $\alpha$ is assumed to be random stochastic variable in time and space within a minimum ($\alpha_{min}$) and maximum ($\alpha_{max}$) value, based on the type of sensor and citizen accuracy. Table 2 summaries the values for the accuracy level that are used in this study and are assumed under the following considerations:

- For both StSc and DySc sensors $\alpha$ values are higher than StPh sensors due to the additional sources of uncertainty introduced with the CS $W_L$ estimation and the consequent conversion to discharge. Moreover, the coefficient $\alpha$ for both StSc and DySc sensors is considered to be a random stochastic variable uniformly distributed in time and space (see Table 2).

- In case of CS observations derived from StSc sensors, $\alpha_{min}$ and $\alpha_{max}$ are assumed to be equal to 0.1 and 0.3 respectively (Mazzoleni et al., 2017a). Accuracy $\alpha$ values mainly account for the uncertainty introduced in the streamflow estimation from $W_L$ by means of the available rating curve derived during the installation of the sensor/staff gauge. The minimum value of $\alpha$ equal to 0.1 assumes a low observational error similar to the one of StPh sensors. The maximum value of $\alpha$ equal to 0.3 assumes a high observational errors in consistency with values used in previous studies (Mazzoleni et al., 2015; Mazzoleni et al., 2017a).

- In case of DySc sensors, the minimum and maximum values are set to 0.2 and 0.5 respectively, i.e. two and five times higher than the uncertainty coming from the StPh sensors. The minimum $\alpha$ equal to 0.2 assumes that $W_L$ can be better estimated from StSc (i.e. by citizens using a reference staff gauge) as compared to the DySc sensors. As described in Lüthi et al. (2014), flow in open channel can be estimated using mobile application only if the channel geometry in known. The maximum $\alpha$ equal to 0.5 is almost double than in case of StSc considering the increasing uncertainty on the assessment of the $W_L$ is due to the limited knowledge of the cross-section geometry at any location.

**Table 2. Assumptions behind the observational errors (based on Weerts and El Serafy, 2006, Rakovec et al. 2012, and Mazzoleni et al. 2017a) according to the sensor type used in this study**

| Sensor type | Assumed accuracy level | Coefficient $\alpha$ | Temporal and spatial variability |
|---|---|---|---|
| Static Physical (StPh) | High | $\alpha = 0.1$ | Fixed location Constant in time |
| Static Social (StSc) | Medium | $\alpha = U(0.1, 0.3)$ | Fixed location Intermittent arrival |
| Dynamic Social (DySc) | Low | $\alpha = U(0.2, 0.5)$ | Variable location Intermittent arrival |

Unfortunately, we do not have any real CS to test the appropriateness of choosing these coefficients' values. A statistical modelling of systematic error against series of CS observations is proposed by Bird et al. (2014). Walker et al (2016) proposes

correlations for consistency of CS with $W_L$ values and rainfall series from nearby hydrologically similar catchments. In addition, to maintain accuracy levels within assumed ranges, Kosmala, et al. (2016) suggest to develop on methods and tools to boost data accuracy and account for bias, include iterative evaluation of CS observations, volunteer training and testing, expert validation and replication across volunteers.

## 4 Experimental setup

In this section, two sets of experiments are performed to test the benefits of assimilation of real-time CS, from a network of heterogeneous static and dynamic social sensors, under different assumptions of CIL.

A 3-day rainfall forecast is used to assess the simulated $W_L$ values along the Bacchiglione River and at the prediction point of

PA.

$W_L$ observations from StPh sensors are assimilated at an hourly frequency, while CS observations from StSc and DySc sensors are assimilated at different intermittent moments to account for the random temporal nature of such observations. The observed and forecasted $W_L$ values are compared at the outlet section of PA.

The number of observations used in each experiment varies based on CIL. Considering a 48h flood event and hourly model

time step, an involvement equal to 1 corresponds to 48 available observations, while with involvement of 0.5 only 24 observations (randomly distributed in time and space) are assimilated.

In addition, several model runs (100) are performed to account for the random accuracy and involvement level in time and space of the citizen in providing CS observations. In each run, a specific $\alpha$ value and arrival moment for each observation are considered and the corresponding $N_{SE}$ value is estimated. From the 100 samples of these $N_{SE}$ values, the corresponding mean

$\mu(N_{SE})$ and standard deviation $\sigma(N_{SE})$ are calculated.

The widely used measure in hydrology, the Nash-Sutcliffe Efficiency ($N_{SE}$) index (Nash 1970), is used to compare simulated and observed quantities:

$$N_{SE} = 1 - \frac{\sum_{t=1}^{T}\left(W_{L,t}^{m} - W_{L,t}^{o}\right)^2}{\sum_{t=1}^{T}\left(W_{L,t}^{m} - \overline{W_{L,t}^{o}}\right)^2} \tag{13}$$

where the superscripts $m$ and $o$ indicate the simulated and observed values of $W_L$, while $\overline{W_L}$ is the average observed water

level. An $N_{SE}$ of 1 represents a perfect model simulation whereas an $N_{SE}$ smaller than zero indicates that the model simulating streamflow is only as skilful as the mean of observed water level. $N_{SE}$ values between 0.0 and 1.0 are generally considered as acceptable levels of model performance (Moriasi et al. 2007).

### 4.1 Experiment 1: Random citizen involvement levels

In the first experiment, CS observations are taken from StSc (Experiment 1.1) and DySc (Experiment 1.2) according to random CIL. Such involvement, closely related to the intermittent nature of the $W_L$ observations, can be considered as the probability to receive an observation at a given model time step. This means that in the case of CIL=0.4 there is 40% of probability to obtain an observation at a given model time step. In fact, in the case of CIL=0, no observation is assimilated and the semi-distributed model is ran without any update, whereas if CIL=1, observations are available at every time step and this situation is analogous to the observation from StPh sensors, which are assumed to be regular in time.

#### 4.1.1 Experiment 1.1: Assimilation of data from static social (StSc) sensors

Experiment 1.1 considers only the assimilation of $W_L$ observations from StSc sensors. The sensors StSc1, 2 and 6, are located in sub-catchment A, B and C respectively, while the other sensors are located along the river reaches of the Bacchiglione catchment (see Figure 1). In contrast to the observations from StPh sensors, the ones from StSc are not regular in time since they are strictly related to the citizen involvement level.

Observation error is defined as in section 3.3 using Eq.(12). The value of $\alpha$ for each StSc sensor is only a function of time $t$ since the location of the sensor is assigned and fixed. Assimilation of $W_L$ observations for different combinations of sensor availability in the different sub-catchments and river reaches is performed.

#### 4.1.2 Experiment 1.2: Assimilation of data from dynamic social (DySc) sensors

In Experiment 1.2, the assimilation of $W_L$ observations coming only from DySc sensors is considered. The two main differences between StSc and DySc sensors are that: 1) DySc sensor locations vary every time step along the river reaches in contrast to StSc sensors whose locations are considered constant in time. In fact, in the case of DySc sensors, the mobile sensor might provide observations in different random places due to the fact that there is no need for a static reference tool to measure the $W_L$; 2) uncertainty in the observations provided by DySc sensors is higher than for those from StSc sensors. This is because for a person it might be difficult to estimate the $W_L$ in a river without any reference device as in the case of StSc sensors.

Analysis on the effect of biased CS observations from DySc sensors is carried out within this experiment. In fact, due to the Bacchiglione catchment complexity and the low available data, the semi-distributed model used in this study may not properly represent internal states away from the calibration point. Consequently, synthetic CS observations may not fully mimic real CS observations, as underlined in Viero (2017). This means that real CSD may be likely biased with respect to the synthetic CS observations generated in this study. For this reason, in case of CS observations derived using DySc sensors, a systematic error is also accounted by means of different values of observations bias:

$$W_{L,t}^{synth} = W_{L,t}^{true} + \gamma_t = W_{L,t}^{true} + W_{L,t}^{true} \cdot U(\gamma_{\min}, \gamma_{\max}) \tag{14}$$

where $\gamma$ is a random stochastic variable function of time, having minimum and maximum values $\gamma_{min}$ and $\gamma_{max}$. In case of no bias $\gamma_{min} = \gamma_{max} = 0$, if $W_L$ is underestimated $\gamma < 0$ and if $W_L$ is overestimated then $\gamma_{max} > 0$. Bias in CS observations from StSc sensors is not considered in this study.

**Table 3. Minimum and maximum values $\gamma_{min}$ and $\gamma_{max}$ in case of 4 different cases of observation bias used in experiment 1.2 and 2**

|  | $\gamma_{min}$ | $\gamma_{max}$ |
|---|---|---|
| **Bias 1 ($\gamma_1$)** | 0 | 0 |
| **Bias 2 ($\gamma_2$)** | -0.3 | 0.3 |
| **Bias 3 ($\gamma_3$)** | -0.3 | 0 |
| **Bias 4 ($\gamma_4$)** | 0 | 0.3 |

The coefficients $\gamma$ are subjectively assumed. In fact, we do not want to argue that a particular value (e.g. 0.3 as in this experiment) should be considered as the default value to estimate bias in real-life crowdsourced observations. Such bias has to be defined based on field experiments with volunteers proving water level observations during real flood conditions. The main point of this analysis is to assess the model sensitivity for different subjective values of $\gamma$. The value of $\gamma$ should be also

defined based on field experiments with volunteers.

**4.2 Experiment 2: Theoretical scenarios of citizen involvement levels**

In this experiment, all the StPh, StSc and DySc sensors are considered. One main problem in citizen science is understanding the motivations that drive citizens to be involved in such activities (Gharesifard and Wehn, 2016). For this reason, a theoretical

assumption about citizen involvement based on their motivations, varying in time and space, is introduced. In the previous experiments, involvement is considered to be random varying from 0 to 1. In this experiment, involvement level is assumed to be a function of the spatial distribution of the population within the Bacchiglione catchment.

As stated by Gharesifard and Wehn (2016), we acknowledge that stronger motivations or intentions are not only driven by a combination of more positive and favourable attitudes. The motivations also rely on stronger positive social pressure and

greater perceived control or self-sufficiency about the means to provide CS observations. Authors further recognized that such rational choices may not apply in case of emergency situations. In this paper, the distinction between favourable attitudes are treated from a theoretical point of view since during the WSI Project no consistent analysis of motivational structures was undertaken for the Bacchiglione case study. Based on Batson et al. (2002), we assume the three main motivations for citizens involvement in collecting data: 1) for their own personal purposes (usefulness of the collected data for personal interest or

direct flood risk management impact); 2) belonging to a community of peers with shared interested; and 3) altruism (beneficing society at large). In order to assess citizen involvement, we propose 3-steps procedure including: 1) estimation of citizen active area; 2) number of active citizens and; 3) citizen involvement curve.

**Step1**: Estimation of the citizen "active area". A hypothetical 500-meter buffer around each sub-river reach of 1000m (spatial discretization of the MC model) is used to identify the area in which the active population might provide CS observations using DySc sensors (see Figure 2). It is assumed that the citizens located further than 500m from the river are not contributing to the collection of CS observations. In the case of the StSc sensor, we assume the active area to be a circle with 500m radius

with the sensor at the centre. Different extents of the buffer will lead to different coverages of the active area, with significant effects on the simulated number of hypothetical involved citizens. However, analyse the implications of different buffer extent on the number of active and consequent flood prediction is out of the scope of this research. Land cover maps are used to identify the main urban area from which citizens might provide CS observations of $W_L$ within the buffer previously estimated (see Figure 2).

**Step 2:** Estimation of the active citizens number. The population density for the different municipalities along the different river reaches is used to estimate the number of citizens within the 500m buffer of each sub-river reach in which the urban areas are located. In the case of agricultural areas, an involvement value equal to zero is considered. In addition, not all citizens would be able to provide CS observations because only a proportion of them uses mobile phones. According to Statistica (2016), the mobile phone penetration in Italy in 2013, the year of the flood event analysed in this study, was about 41%, which

means that about 41% of the population was potentially able to submit data. In view of the lack of a better source, we assume that this proportion is valid also for the regional scope. Therefore, to estimate the potential number of active citizens that could submit data close to the river reach, we first estimate the total population enclosed in a cell of 1km long by 1km wide (a buffer of 500m from each side of the river), and then estimate the 41% of them. Table 4 summarizes the results for the case of the StSc sensors and Table 5 those for the DySc sensors. In Table 5, the active citizens are divided by the number of sub-reaches

(3 for reach 6). For reach 6 (km 3-4-5), main urban areas are contained in more than one sub-reach. Naturally, for a better estimation of these values, a more exhaustive social-economic analysis should be performed.

**Table 4. Estimate of the active population that potentially can provide CS observation of $W_L$ from StSc sensors**

| Sensor | Municipality | Active area (m²) | Density (inhab/km²) | Population (inhab) | Active citizens (inhab) |
|---|---|---|---|---|---|
| **StSc–1** | Schio | 206828 | 597 | 124 | 51 |
| **StSc–2** | | 71293 | | 43 | 18 |
| **StSc–3** | Malo | 100734 | 491. | 50 | 21 |
| **StSc–4** | Villaverla | 359744 | 400 | 144 | 59 |
| **StSc–5** | Caldogno | 67311 | 720 | 49 | 20 |
| **StSc–6** | Costabissara | 421778 | 563 | 238 | 98 |
| **StSc–7** | Vicenza | 86544 | 1400 | 122 | 50 |
| **StSc–8** | | 241451 | | 339 | 139 |
| **StSc–9** | | 415513 | | 583 | 239 |
| **StSc–10** | | 500000 | | 700 | 287 |

**Table 5. Estimate of the active population that potentially can provide CS observation of $W_L$ from DySc sensors**

| Reach | Municipality | Active area (m²) | Density (inhab/km²) | Population (inhab) | Active citizens (inhab) |
|---|---|---|---|---|---|
| 1 (km6-7-8) | Marano Vicentino | 608985 | 800 | 487 | 200 |
| 2 (km2) | Schio | 39536 | 597 | 24 | 10 |
| 3(km8) | Villaverla | 359744 | 400 | 144 | 59 |
| 3(km11) | Caldogno | 232474.1 | 720 | 167 | 69 |
| 4(km2) | Dueville | 30692 | 701 | 22 | 9 |
| 4(km3) | Caldogno | 191988 | 720 | 138 | 57 |
| 4(km5) | | 292519.8 | | 211 | 86 |
| 5(km1) | Costabissara | 351921 | 562 | 198 | 81 |
| 5(km2) | | 119898 | | 67 | 28 |
| 5(km3-4-5) | | 212453 | | 100 | 41 |
| 6(km1-2) | Vicenza | 129816 | 1400 | 90 | 37 |
| 6(km3-4-5) | | 1156964 | | 539 | 221 |

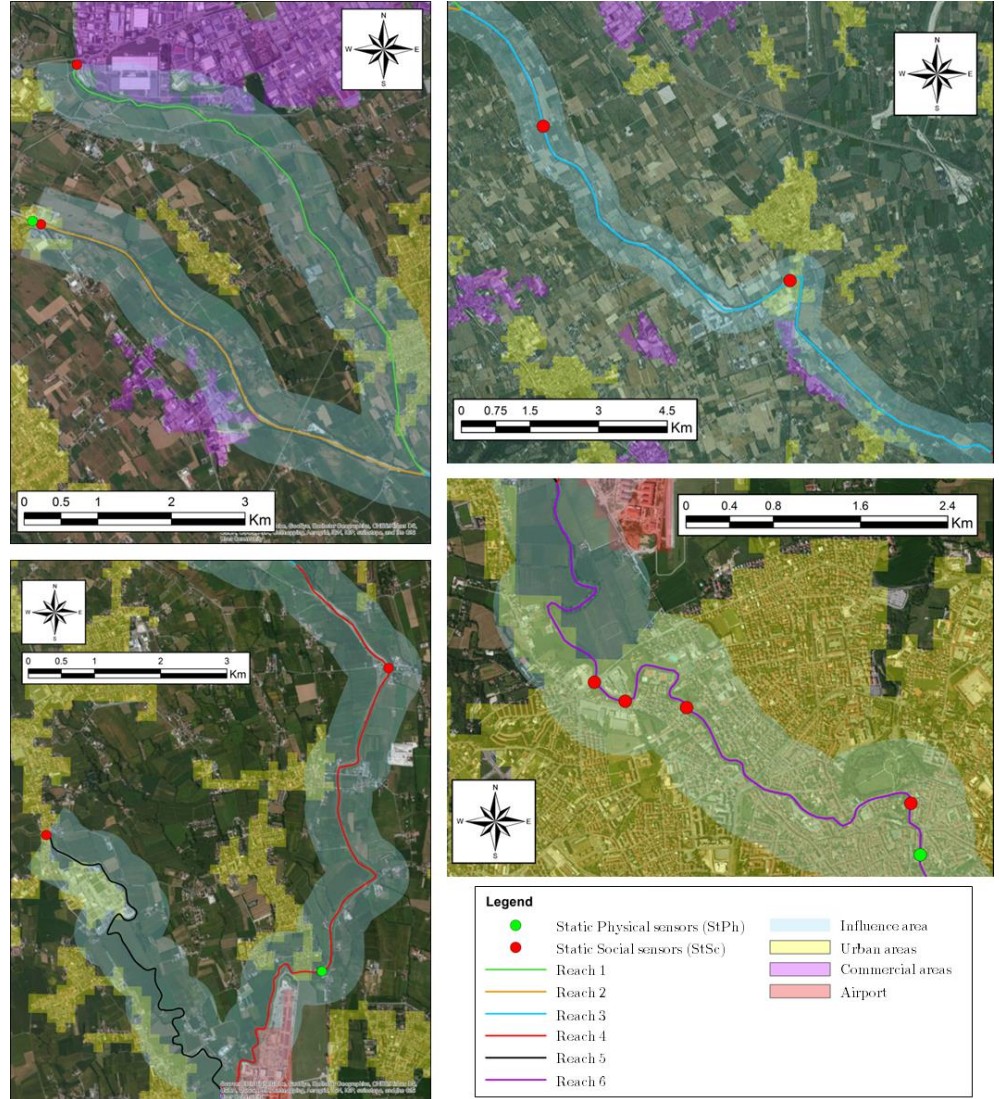

**Figure 2. Representation of the different Bacchiglione river reaches, land use (Corine Land Cover, 2006), location of the StSc and StSc sensors and the 500-meter buffer**

**Step 3**: Estimation of the theoretical citizen involvement curve. It is now necessary to estimate the citizen' level of involvement based on the hypothetical number of active citizens and their motivation for sharing data. For this reason, three different involvement curves, representing each a scenario and corresponding number of active citizens, providing the Maximum Citizen Involvement Level (MCIL) are proposed. These scenarios are based on Batson et al., (2002), whose aggregated categories of citizen's motivations are still in agreement with more comprehensive and detailed analysis such the ones recently reported in Geoghegan et al. (2016) and Gharesifard et al.(2017).

In the scenario 1, we assume that citizens collect data mainly for their own personal use. In this case, the MCIL is low for low number of citizens, while it grows following a logistic function, Eq.(15), for increasing numbers of people.

$$MCEL = \frac{K \cdot P_o \cdot e^{r \cdot Pop}}{K + P_o \cdot \left(e^{r \cdot Pop} - 1\right)} + w \tag{15}$$

Where:

$P_{op}$ is the population number;

$r$ is the growth rate, we assumed two different values of $r$ (0.04 and 0.08);

$K$ is the carrying capacity, i.e. maximum value of MCIL, assumed equal to 1;

w is a coefficient related to the additional CS observations are also driven by societal benefits (third citizen scenario explained below);

$P_o$ is the minimum value of MCIL assumed equal to 0.01.

In the scenario 2, citizens might decide to collect and share CS observations driven by a feeling of belonging to a community of peers with shared interests and vision. In this case, it is assumed that a maximum value of MCIL is achieved for small population values while for increasing population this value is reducing. This scenario follows an inverse logistic function as shown in the graphical representation of scenario 2 in Figure 3.

In the scenario 3, enthusiast individuals might provide additional information driven by moral norms and the wish to create knowledge about the hydrological status of the river, benefiting society at large. This is potentially a much smaller subset of the population. The added value of this information is accounted for in Eq.(15) by means of a coefficient w. Table 6 summarizes the different involvement curves based on the previous scenarios and different values of the coefficients $r$ and $w$.

At the next phase of analysis, a number of model runs (100) are carried out, considering the random values of citizen involvement from 0 to the MCIL according to the given involvement scenarios and the population. For example, considering scenario 1 and 150 inhabitants enclosed in a given river sub-reach, several model runs are performed for involvement values varying from 0 to 0.65 based on Figure 3. In case different CS observations coming at the same time from different sensors, only the most accurate observation, i.e. having the lower value of the coefficient $\alpha$ in Eq.(12), is assimilated in the hydrological and/or hydraulic model. Another approach could be to assimilate all measurements instead of only the most accurate ones. In this case, each observation is used within the assimilation scheme with the account of its error: less weight would be given to the more uncertain observations.

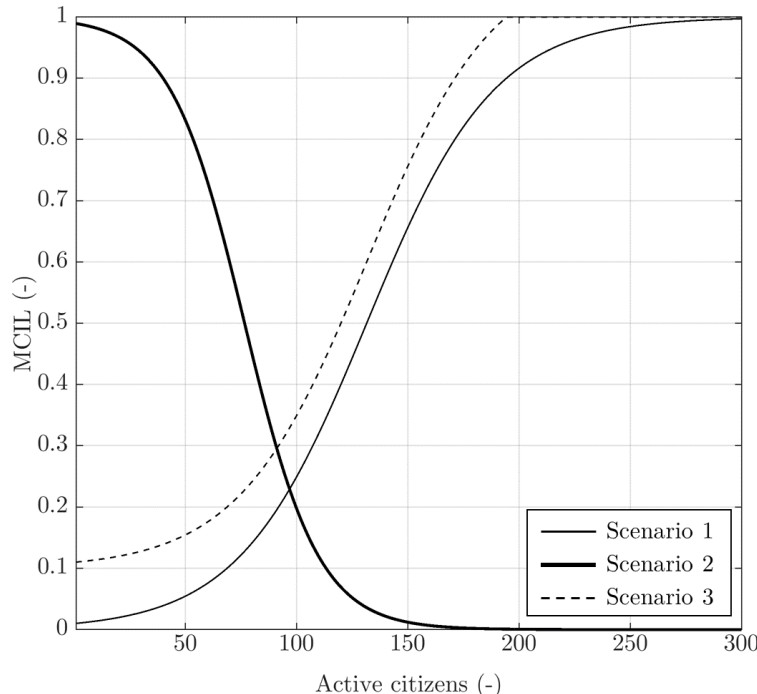

**Figure 3. Representation of the theoretical MCIL scenarios based on number of active citizens.**

**Table 6. Involvement curves based on different citizen motivations**

| Involvement scenario | Citizen motivation | *Growth rate (Factor r in Eq 15)* | *Additional CS observations (Factor w in Eq. 15)\** |
|:---:|:---:|:---:|:---:|
| **1** | Own purposes (1) | 0.035 | 0 |
| **2** | Shared or community interests (2) | 0.060 | 0 |
| **3** | Social benefits(3) | 0.035 | 0.10 |
| | *Increment **applies when** CS are also driven by **societal benefits** (third citizen motivations) | | |

Finally, in this experiment it is also investigated the effect of the spatial variability of smartphone penetration and decrease of citizens involvement levels in time. For this reason, higher (double) percentage of active citizens in Vicenza is assumed (smartphone penetration of 80%), while random values of the coefficient *r* are considered to represent lower involvement levels over time.

# 5 Results

## 5.1 Experiment 1

### 5.1.1 Experiment 1.1

In Experiment 1.1, the effect of different CIL on the assimilation of CS observations from StSc sensors is analysed. Figure 4 aims to represent the $\mu(N_{SE})$ values obtained when assimilating CS observations from StSc sensors located in a different sub-catchments (hydrological model) and river reaches (hydraulic model) for a 1-hour lead time. For example, in Figure 4.a, the $N_{SE}$ values obtained assimilating CS observations from sub-catchments A and river reach 3 are shown for different involvement values.

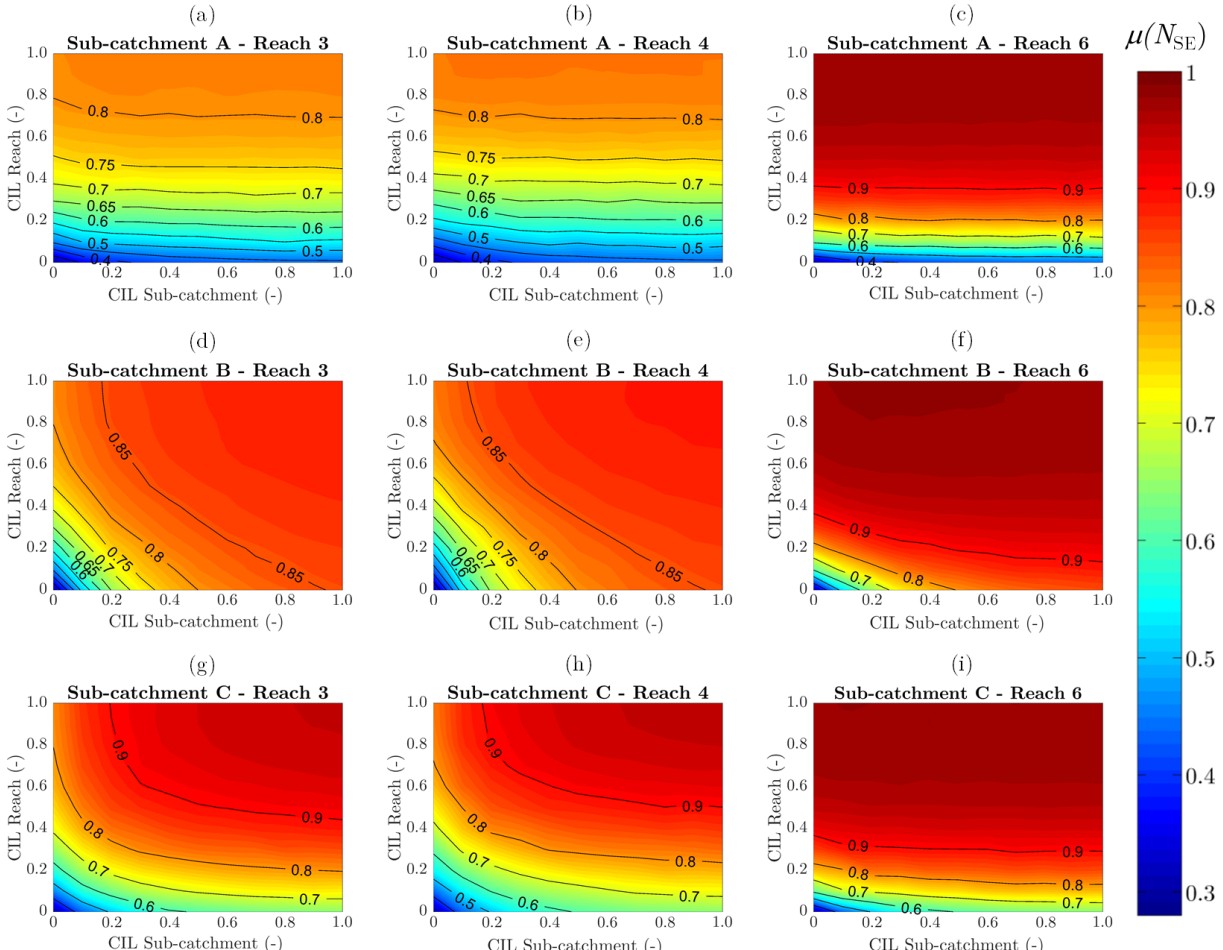

**Figure 4.** $\mu(N_{SE})$ **values obtained assimilating CS observations from a combination of StSc sensors located in different sub-catchments and river reaches with 1-hour lead time in case of different CIL values.**

Figure 4 shows that $N_{SE}$ values are less affected by the assimilation of CS observations located in the sub-catchment A than in the other reaches. In fact, Figure 4a, b and c, it is clear that $N_{SE}$ values change only for different involvement values of StSc sensors along reach 3, 4 and 6, while constant $N_{SE}$ values are achieved for varying involvement values of the StSc (sub-catchment A). As previously shown, for a low lead time value, $N_{SE}$ is higher in case of StSc sensors located in reach 6 rather than in the other river reaches 3 and 4.

In case of assimilation in sub-catchment B, Figure 4d, e and f, higher $N_{SE}$ values are achieved if compared to the ones for the sub-catchment A (first row of the same figure). In particular, $N_{SE}$ values are mainly influenced by different involvement levels of CS observations from sub-catchment B than from river reaches 3. However, moving from upstream (reach 3) to downstream (reach 6) a switch in the model behaviour can be observed, with an increasing influence of involvement in StSc sensors located in the river reach close to the PA station, as previously demonstrated (see contour map of sub-catchment B and reach 6 in Figure 4).

Similar results are shown for StSc sensors located in sub-catchment C and different river reaches, Figure 4g, h and i. However, involvement levels in upstream river reaches affect the $N_{SE}$ values more than the involvement of StSc sensors in sub-catchment C. The same behaviour is manifested considering StSc sensors located from upstream river reach to downstream. The third row of Figure 4 can be considered as an average situation between the first (sub-catchment A) and the second (sub-catchment B) row of the same figure.

Figure 5 is analogous to Figure 4, but with a lead time of 4 hours. Overall, as expected, the $N_{SE}$ values are lower for lead time of 4 hours, if compared to that of 1 hour. Model results are dominated by the assimilation in the sub-catchments A, B and C if compared to the involvement in reaches 4 and 6. This is due to the fact that assimilation from the hydrological model allows achieving good model predictions in case of high lead values. An intermediate situation is achieved for reach 3. It can be seen that assimilation of CS observations in this upstream river reach allows to obtain higher $N_{SE}$ values in case of high lead times due to the longer travel time than the one of StSc sensors located closer to PA (e.g. reach 6). Citizen involvement in reach 3 affects the $N_{SE}$ values more than the involvement levels in sub-catchment A and C. Moreover, as in case of Figure 4 for 1-hour lead time, involvement in sub-catchment B have higher impact on $N_{SE}$ values than involvement in reach 3. A more detailed analysis on the effect of sensor location and lead time is provided in Mazzoleni et al. (2017a).

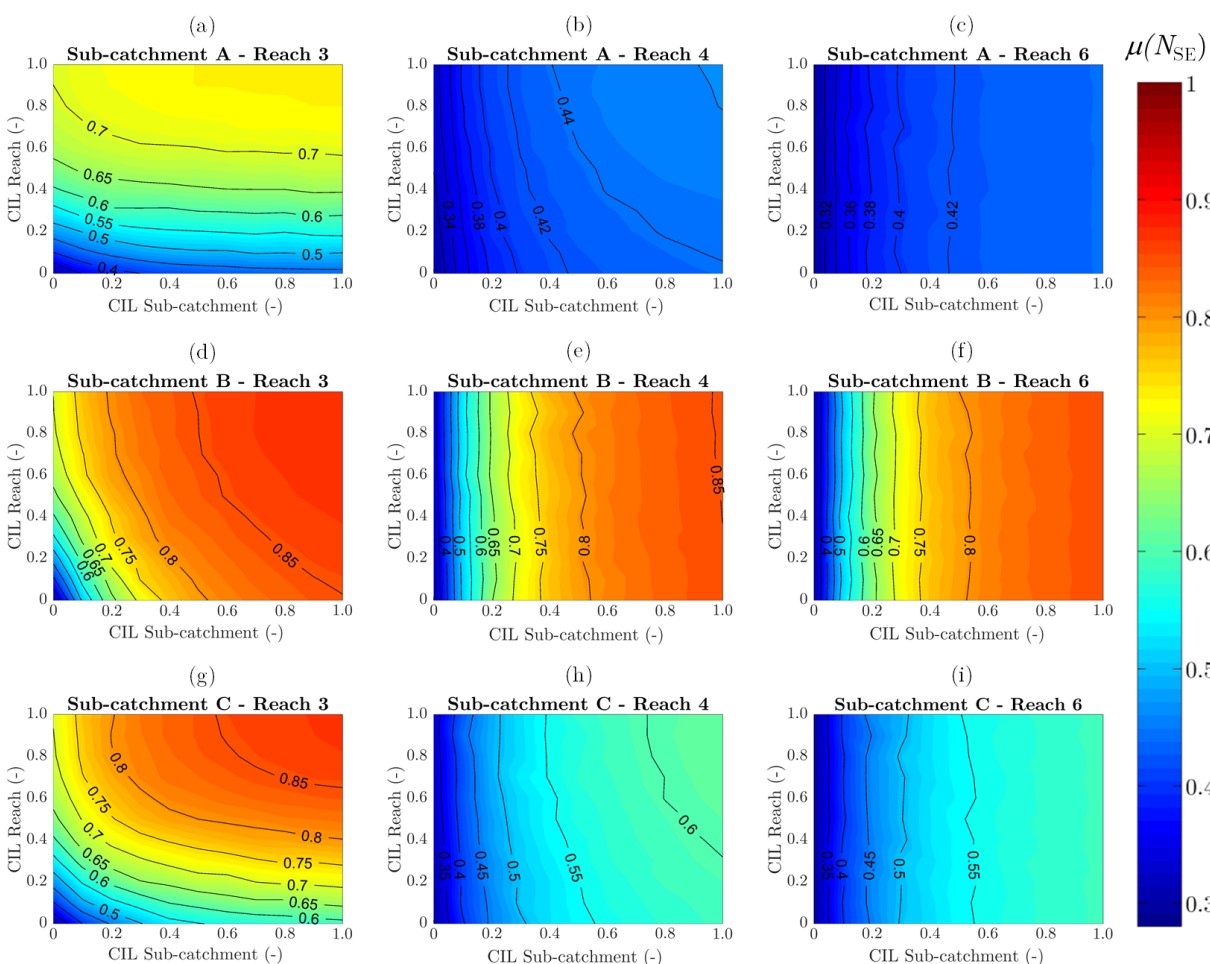

**Figure 5.** $\mu(N_{SE})$ **values obtained assimilating CS observations from a combination of StSc sensors located in different sub-catchments and river reaches with 4-hours lead time in case of different CIL values.**

### 5.1.2 Experiment 1.2

In Experiment 1.2, the effect of CIL in assimilating CS observations only from DySc sensors is analysed. In this case, the DySc sensors are assumed to be located only along the river reach 3, 4 and 6 so only the hydraulic model is used in this experiment. Also in this experiment, 100 runs are carried out to account for the random accuracy and location of the CS observations.

In Figure 6, DySc sensors are assumed to be present every 1000m, while CIL changes in each model run. This means that CS observations that are available at one time step at one specific location may not be available at the same location for the next time steps. It can be observed that in most of the cases $\mu(N_{SE})$ values converge asymptotically to some threshold, as involvement level increases. Among the three river reaches, 3 and 4 are the ones providing higher $N_{SE}$ values for low involvement levels. This can be related to the high number of DySc sensors located in reach 3 (13 sensors) and 4 (8 sensors).

Although, reach 6 is better performing in case of high involvement levels, high $\sigma(N_{SE})$ values are obtained for this reach, showing a significant sensitivity of model performance in case of different CIL in the hydraulic model. Assimilating CS observations from DySc sensors at different reaches induces an overall improvement of $\mu(N_{SE})$ and reduction of $\sigma(N_{SE})$. Lowest $\sigma(N_{SE})$ values are obtained including DySc sensors from reaches 3 and 4. However, this reduction in the $\sigma(N_{SE})$ values does not correspond to a higher improvement in $\mu(N_{SE})$. In fact, the highest $\mu(N_{SE})$ are achieved joining sensors from reach 4 and 6, i.e. the closest river reaches to the PA station. Similar results in terms of $\mu(N_{SE})$ and $\sigma(N_{SE})$ are obtained joining reaches 3 and 6.

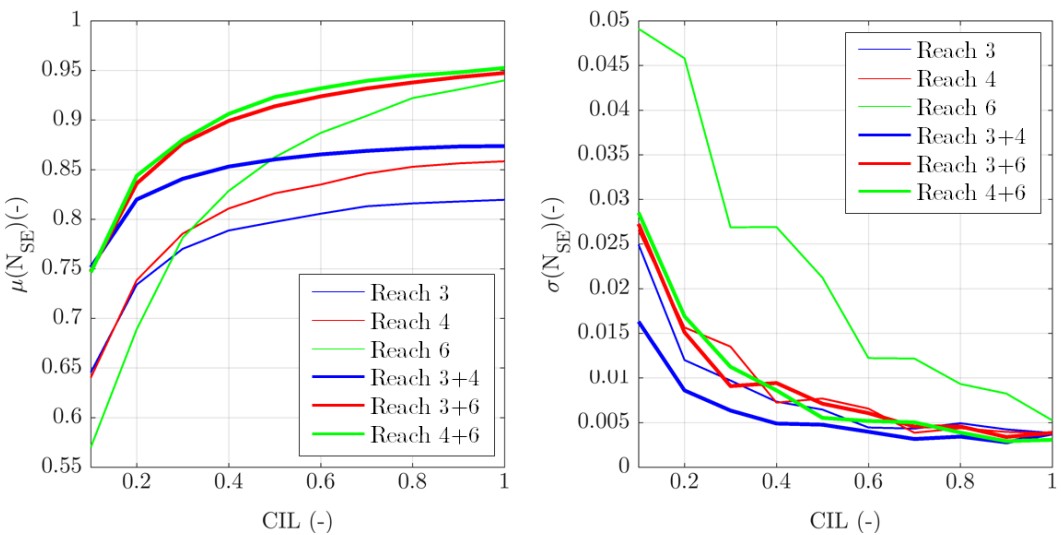

**Figure 6. Effect of different levels of involvement, in terms of $\mu(N_{SE})$ and $\sigma(N_{SE})$ in the assimilation of CS observations from DySc sensors for different CIL values**

It is worth noting that in Figure 6, no bias in the observations from DySc sensors is considered.

Figure 7 presents the $\mu(N_{SE})$ values obtained considering random locations of DySc sensors along the river reaches 3, 4 and 6 in 4 different cases of CS observation bias for 1 hour lead time. As reach 6 has five different sub-reaches of 1000m, CS observations from only five sensors can be assimilated. However, in Figure 7 a total number of 13 DySc sensors is considered. In these experiments, location of DySc sensors are randomly generated. It might happen that two sensors are located, say, at distances of 2600m and 2900m from the upstream boundary condition. Because of the small spatial discretization of the hydraulic model (1000m), it is assumed that the difference between the hydrographs estimated between the two different model discretization is negligible. For this reason, the two CS observations from the DySc sensors at 2600m and 2900m are simultaneously assimilated at the third sub-reach. In this way, it is possible to assimilate CS observations from a number of DySc sensors higher than the number of model spatial discretization points.

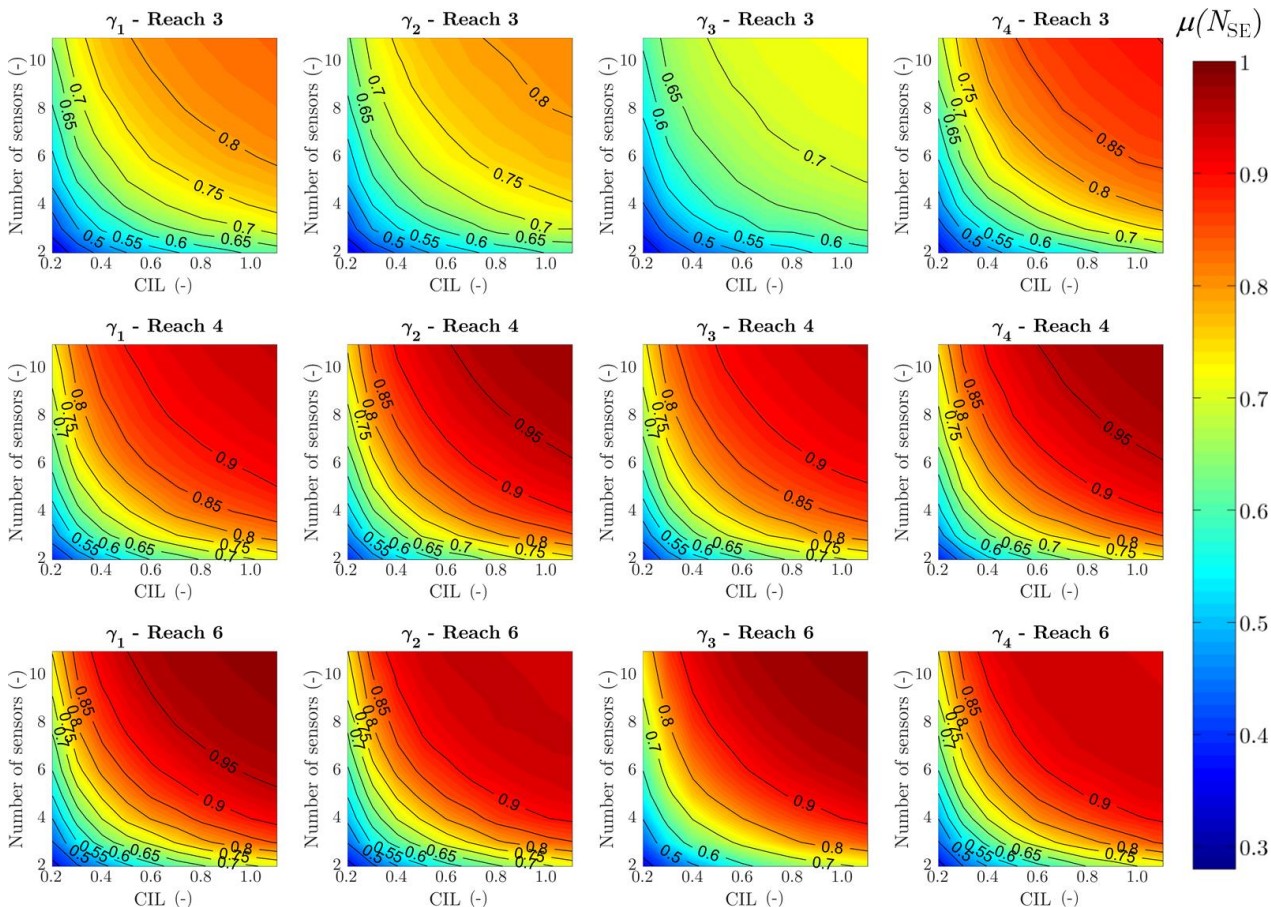

**Figure 7.** $\mu(N_{SE})$ **values obtained considering random location of dynamic social (DySc) sensors along the river reaches 3, 4 and 6 in 4 different cases of CS observation bias for 1hour lead time and Citizen Involvement Level (CIL) values**

As it can be observed, different $\gamma$ values (bias assumptions) affect the model performance in different ways. Underestimation of the CS observations ($\gamma_3$) induces a reduction of the $\mu(N_{SE})$ values due to the underestimated forecasted precipitation. In consequence the underestimation of water level hydrograph at PA in case of no model update. For the same reason, overestimation of CS observations ($\gamma_4$) causes an increase in model performance especially for a low number of DySc sensors and involvement levels. In case of $\gamma_2$ the behaviour in-between $\gamma_3$ and $\gamma_4$ can be observed.

**5.2 Experiment 2**

Experiment 2 focuses on the assimilation of CS observations from a distributed network of heterogeneous StPh, StSc and DySc sensors. In particular, the involvement level is calculated in a more realistic way accounting for the population living in the range of 500m from the river. Based on Figure 3, different MCIL values are calculated for the three scenarios in collecting and sharing $W_L$ observations. It is worth noting that Bias 2 is considered in the CS observations from DySc sensors.

Figure 8 shows $\mu(N_{SE})$ values in case of different involvement scenarios and MCIL according to the different type of sensors. A random value of involvement level between 0 and MCIL is considered for a given river sub-reach and model run. In particular, in Figure 8, smaller values of MCIL such as MCIL1, MCIL2, MCIL3, MCIL4 and MCIL5 are estimated as to 0.2 MCIL, 0.4 MCIL, 0.6 MCIL, 0.8 MCIL and MCIL, respectively. It can be noticed that scenario 2 is the one providing the best model improvements, followed by scenario 3. Involving the enthusiastic people (scenario 3) helps to improve $\mu(N_{SE})$, especially for low involvement values. Scenario 1 is the one that gives the lowest $\mu(N_{SE})$ values due to the lowest growth rate of the involvement curve and consequent lower involvement of citizens.

In scenarios 1 and 3, the steepest vertical gradient of the contour plot can be observed, leading to the conclusion that model results seem to be more sensitive to the change of MCIL values in StSc sensors rather than DySc sensors. However, the gradient reduces with scenario 2.

In the previous analysis, $N_{SE}$ is used as the only performance indicator without considering improvement in the prediction in the peak and rising limb of the hydrograph, which are extremely important in case of operational flood management. For this reason, the relative error between the observed streamflow peak and simulated peak (see Eq. 16) is included to better assess the assimilation of crowdsourced observations from an operational point of view.

$$E_{RR} = \frac{\left(W_{L,P}^{O} - W_{L,P}^{S}\right)}{W_{L,P}^{O}} \tag{16}$$

where $W_{L,P}^{O}$ and $W_{L,P}^{S}$ are the observed and simulated streamflow ($m^3 s^{-1}$). The results reported in Figure 8 shows comparable results to the ones achieved using $N_{SE}$. Including CS observations from enthusiast citizens seems not to lead to a more accurate representation of the peak discharge. In fact, similar $\mu(N_{SE})$ values are achieved between scenario 1 and 3. However, error in peak prediction is lower in scenario 1 than in scenario 2. It can be observed that $E_{RR}$ values are clearly more sensitive to the different involvement values in StSc sensors than from the DySc ones (vertical gradient).

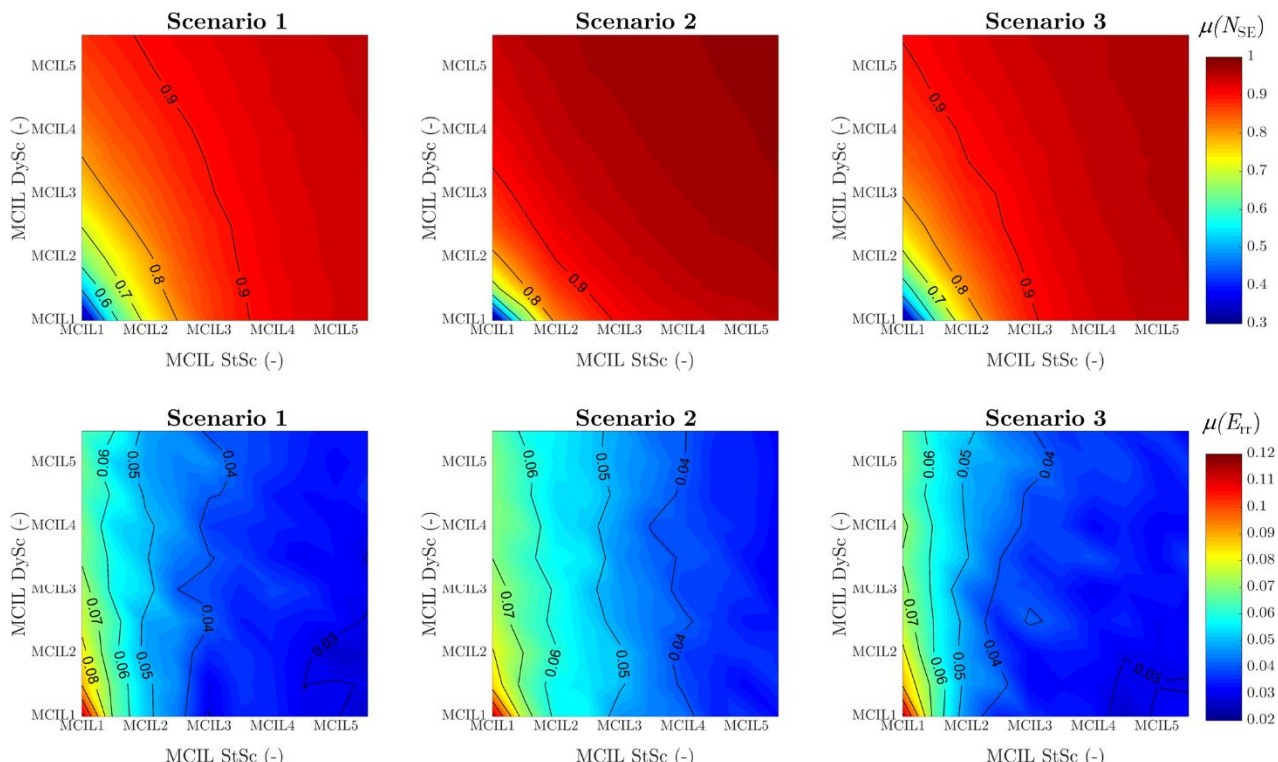

**Figure 8. $\mu(N_{SE})$ and $\mu(E_{rr})$ values obtained in case of different Maximum Citizen Involvement Level (MCIL) scenarios comparing involvement level from StSc and DySc sensors**

In the previous analysis, unrealistically high citizen involvement (up to 80%) is considered. For this reason, the following analysis focuses more into the lower part of the theoretical involvement curve, assuming more realistic CIL. In particular, the maximum carrying capacity of the logistic curve (K) is changed from 0.01 up to 1. In case of K equal to 1, the values of $\mu(N_{SE})$ related to the different scenarios are estimated as mean average of the contour plot showed in Figure 8. The same analysis is performed for the vector of different values of K.

The results of this analysis show an expected reduction in the model performances for low values of the parameter K (which indicates the maximum possible level of involvement). It can be noted that if K is equal to 0.5, assimilation of crowdsourced observations still provide significant model improvement for all the different scenarios even though the involvement is halved. As expected, $\sigma(N_{SE})$ values tend to increase for low involvement of citizens. From Figure 9, it can be seen that $\mu(N_{SE})$ values do not follow a linear trend as expected. On the contrary, it tends to drop for values of K between 0 and 0.2 (for example in scenario 3), while for higher K values the $\mu(N_{SE})$ does not grow significantly. In particular, for K values higher than 0.5, scenario 2 provides the highest $\mu(N_{SE})$ values. Besides, for lower K values than 0.5, scenario 3 is the one leading to better model performances. This is because the presence of enthusiast individuals keeps high involvement values even for low values of K. Regarding the variability of $N_{SE}$, i.e. $\sigma(N_{SE})$, for values of K lower than 0.4, high $\sigma(N_{SE})$ can be observed in scenario 1.

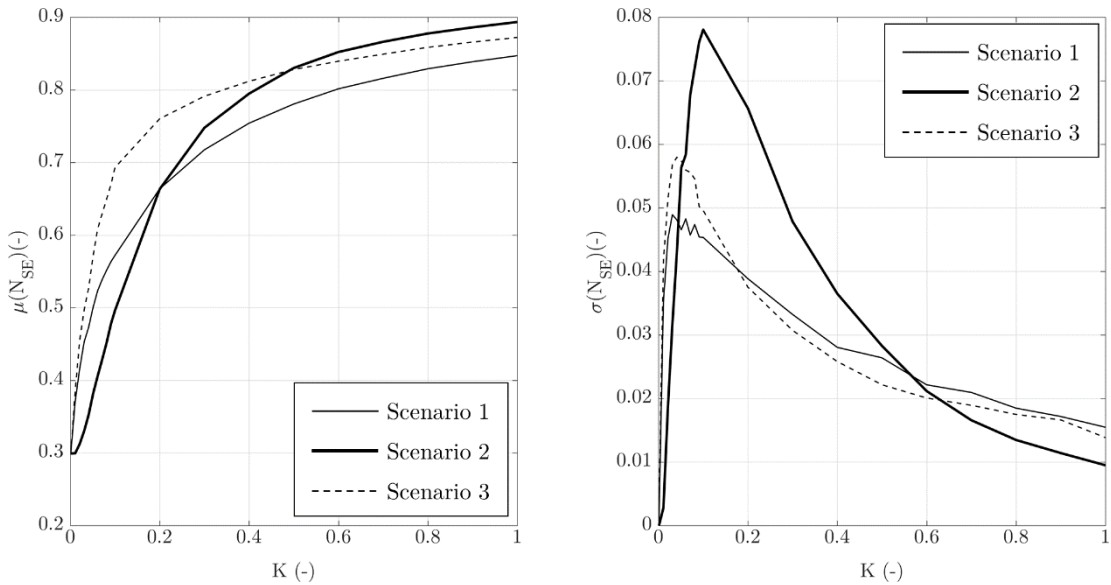

**Figure 9.** *$\mu(N_{SE})$ and $\sigma(N_{SE})$ values obtained considering varying values of K for different involvement scenarios*

Additional analysis considering negative and positive bias (Bias 3 and 4 in Table 3) in the CS are considered. As expected, it can be observed that Bias 4 provides higher $N_{SE}$ values than Bias 2 since model without update underestimate observed streamflow/water level. Moreover, results obtained using observations with Bias 3 have lower $N_{SE}$ than the results with Bias 2. However, in both Bias 3 and 4, such changes in $N_{SE}$ are very small, leading to the conclusion that assimilation of biased $W_L$ observations during the May 2013 flood event in the Bacchiglione River do not reduce model performances.

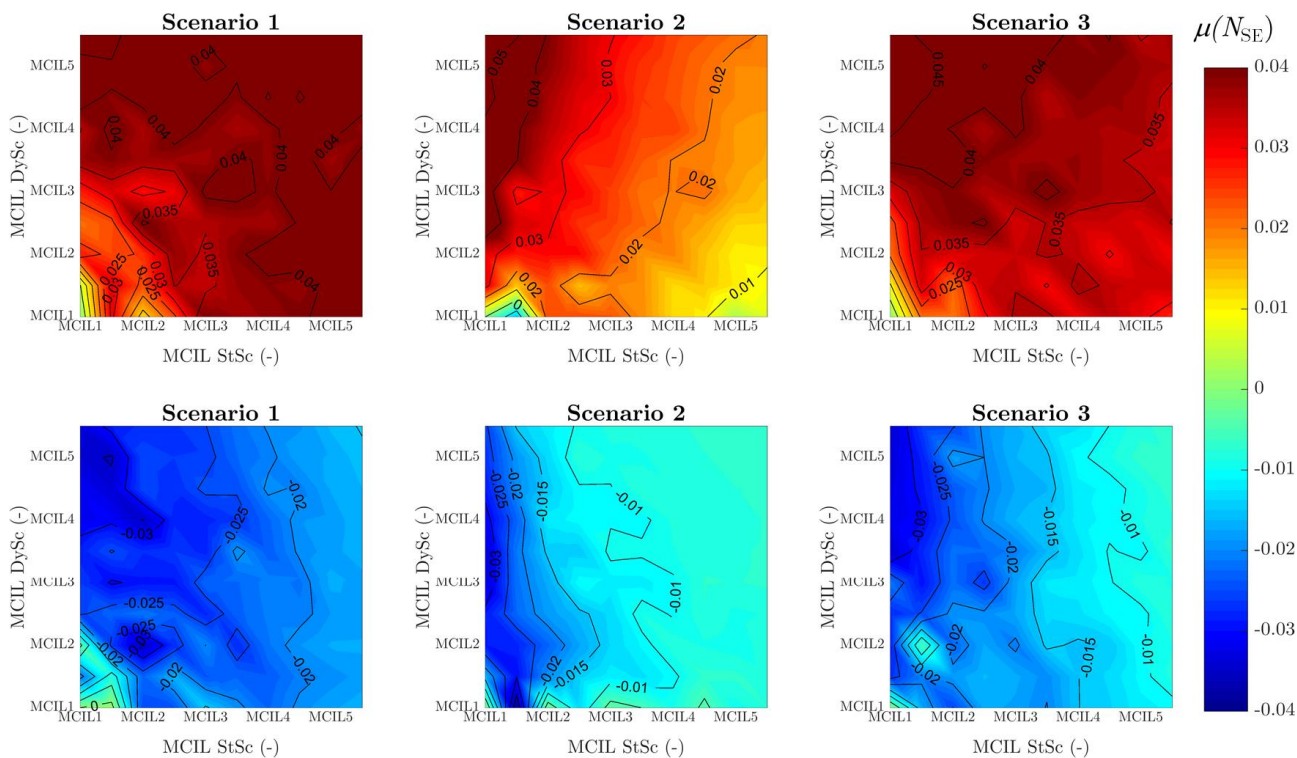

**Figure 10.** *Difference between μ(N_SE) values obtained considering Bias 2 with Bias 3 (first row) and Bias 2 with Bias 4 (second row) for different involvement levels from StSc and DySc sensors*

*Effect of spatial variability of smartphone penetration*

The value of smartphone penetration depends mainly on the geographic area and on the characteristic of the population. We assume that not everyone is prone to use smartphone to collect and share water level data due to their age and habits. However, smartphone penetration and consequent percentage of active citizens may change spatially. In the following simulations, a higher percentage of smartphone users (80%) is assumed in the urbanized area of the municipality of Vicenza. From Figure 11 it can be seen that increasing the smartphone penetration in Vicenza does not affect model results in case of scenario 2.

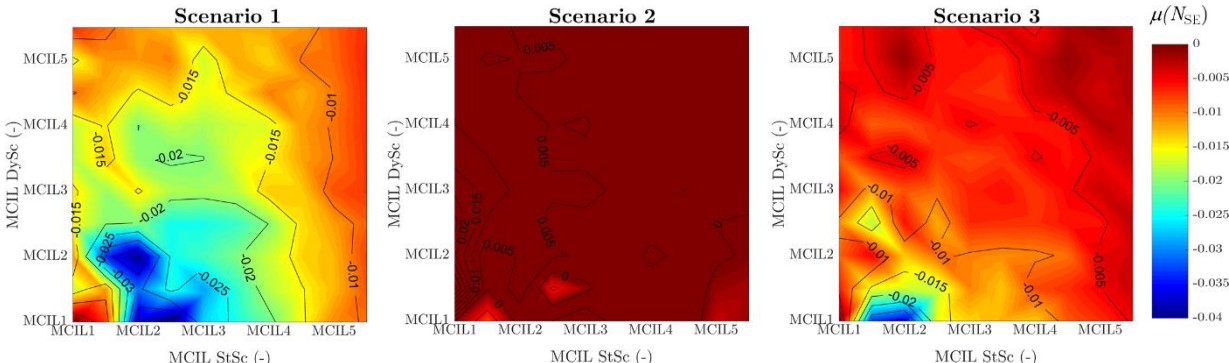

**Figure 11.** *Difference between $\mu(N_{SE})$ values obtained considering standard and higher active citizen percentage in the municipality of Vicenza for different involvement levels from StSc and DySc*

For this scenario, no involvement is assumed in highly urbanized areas such as the municipality of Vicenza. Higher number of smartphones in Vicenza partially affects only scenarios 1 and 3. In these scenarios, an expected increment in the model performance (due to the higher involvement in Vicenza), can be observed. However, small increments in the $N_{SE}$ values are reported in Figure 11, with a maximum difference of 0.04 between normal and higher smartphone penetration.

*Effect of temporal variability of citizen involvement*

In the previous analyses, CIL is considered constant in time. However, in real practice, involvement may decrease if citizens are not properly involved in a water observatory, so for the assimilation of CS observations it is also important to consider also this situation. A possible idea to represent the decrease of involvement level on time could be to assume varying values of growth rate *r* of the logistics curve over time.

In Figure 12, results of sensitivity analysis of model results with respect to the varying values of the coefficient *r* of Eq.15 are presented. Only scenario 3 and three different values of *w* are considered. The results demonstrate that decreasing involvement over time (low values of *r*) lead to a reduction of the model performance and consequently inaccurate flood forecasts. This is an expected result that demonstrates again the importance of keeping citizens continuously engaged. However, such reduction of model performances is significant only for values of *r* lower than 0.3, leading to the conclusion that model performances can still be high even if involvement reduces over time.

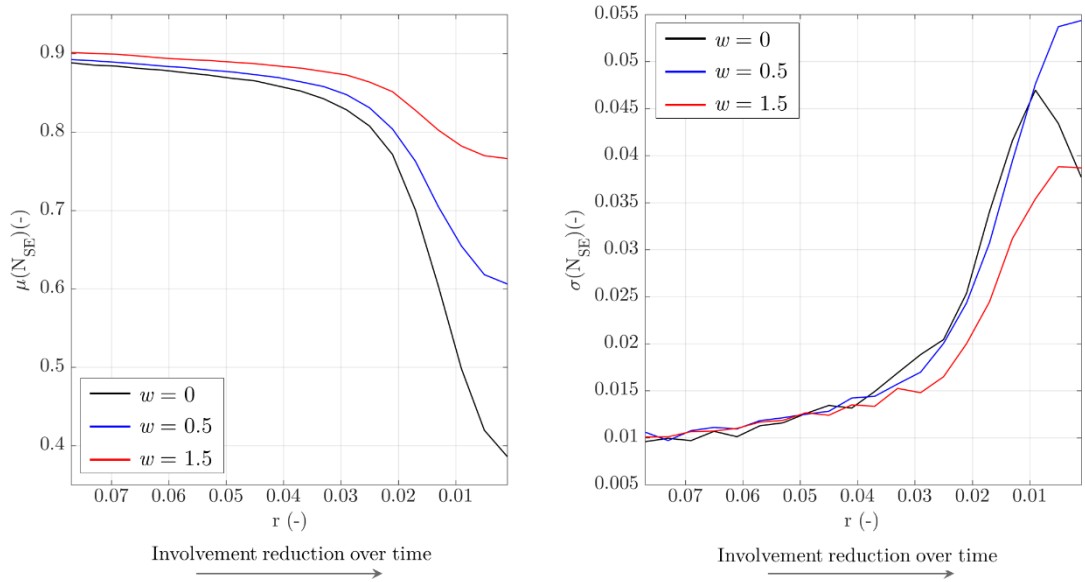

**Figure 12.** *μ(N_{SE}) and σ(N_{SE}) values obtained considering varying values of the coefficient r for scenarios 1 and 3 with three different values of w*

## 6 Discussion

In flood risk management, CS observations of hydrological variables can potentially contribute to the situational awareness and to support decision-making (Howe, 2008; Alfonso, 2010; Rotman et al., 2012; Gura, 2013; Bonney et al., 2014; Buytaert et al., 2014). ICT-enabled citizen observatories become possible via, for example, mobile and web-based easy-to-use sensors and low-cost monitoring technologies (Jonoski et al., 2012). However, the fact that ICT tools and citizen observatories initiatives are in place does not automatically imply a higher level of citizen involvement - due to intermittency and timely

availability of CS observations (De Grossi et al., 2013). This section aims to summarise the main findings of our study and to analyse the pros and cons of using CS observations for improving flood predictions. It is worth noting that in this study we do not refer to how to get the citizens involved but rather to the probability of receiving a CS observation based on the citizen's own interest or intention in collecting water levels. Engagement and involvement levels are related and represent a huge barrier to collect CS observations (Starkey et al., 2017).

Overall, the results we have obtained are in accordance with the recent studies on the use of (real) crowdsourced observation in the area of water resources management (Gaitan et al., 2016; Giuliani et al., 2016; de Vos et al., 2017; Rosser et al., 2017; Scheider et al., 2017; Starkey et al., 2017; Yu et al., 2017). In particular, any improvement of model performance with respect to the current practice for flood forecasting in the catchment used by Alto Adriatico Water Authority with no model update, provides additional useful information for flood risk management. The results from Experiment 1.1 (assimilation only from

StSc) show that model outputs depend on the particular sub-catchment and river reach in which the observations are assimilated. In fact, we also found that accuracy of the assimilation process is highly dependent on different factors including

the total number of observations, their spatial distribution and their accuracy as demonstrated by Schneider et al. (2017) and Starkey et al. (2017) by using real CS observations. In addition, assimilation of CS observations into the hydrological model tends to provide lower improvement than the assimilation in the hydraulic model. However, assimilation in hydrological models ensures a better model prediction for high lead time values than the assimilation in the hydraulic model. This is due to

the high travel time needed to reach the prediction point of PA (around 22 hours from the outlet of sub-catchment B). For operational flood management it is advisable to consider model results in which observations at upstream location of the catchment are assimilated in both hydrological and hydraulic model. In Experiment 1.2, assimilation of CS observations from DySc sensors produced an overall improvement of model performances in terms of $\mu(N_{SE})$ increase and $\sigma(N_{SE})$ reduction. Higher values of $\mu(N_{SE})$ are achieved assimilating CS observations coming from multiple river reaches, in particular, for those

reaches close to PA. Due to the fact that the model without assimilation underestimates the observed water level, overestimated biased CS observations tends to increase model performances. Comparable results were obtained in Rakovec et al. (2012) and Mazzoleni et al. (2015, 2017a) in case of assimilation of distributed sensors in hydrological modelling.

The aim of Experiment 2 is to investigate the effects of different, theoretical, level of involvement in the assimilation of CS observations coming from heterogeneous sensors (StPh, StSc and DySc). Our findings demonstrate that sharing CS

observations driven by feeling of belonging to a community of peers (scenario 2 in the proposed theoretical social model) can help improving flood prediction if such a small community is located upstream of a particular interest point. The results achieved for scenario 1 point out that a growing participation of citizens motivated by personal interests, sharing hydrological observations in big cities, can help improving model performance. In particular, the model results can benefit from the additional observations provided by enthusiastic citizens (scenario 3). Similar conclusions are reported in Starkey et al. (2017)

where it is demonstrate the importance of a proper engagement for providing additional source of catchment information.

Finally, it is important to investigate the effect of varying percentage of smartphone usage in space and decrease of citizen involvement over time. Percentage of active citizens may change spatially in densely populated areas such as the municipality of Vicenza. Increasing the smartphone penetration in Vicenza would not affect model results in case of scenario 2, because no involvement is assumed in densely urbanized areas. High percentage of active citizens in Vicenza affects only scenarios 1 and

3. However, because the number of active citizens in Vicenza is already high for a smartphone usage of 41%, the model improvement is not significant for higher percentage of active citizens. This means that in the proposed theoretical involvement model more active citizens (i.e. more mobiles available) will not significantly improve involvement and affect the model performance. It is worth noting that a more exhaustive social-economic analysis should be performed in order to better define the smartphone penetration and consequent percentage of active citizens.

In this study we assume intrinsic motivation, constant in time, differentiated according to the level of involvement. However, a main challenge in citizen science is to keep this involvement high in the long term. In case of flood events, citizen involvement tends to disappear if no other event will occur in a short time. In fact, depending on the memory of the community, the awareness of flood risk decreases over time (Raaijmakers et al., 2008), and, therefore, the tendency to be engaged in data collection will also reduce or even disappear. For this reason it is important to keep citizens engaged using for example

gamification approaches or periodic meetings/seminars with interested participants. However, the main goal of this paper is not to review or propose approaches to engage and keep citizen involvement over a long time. For this purpose, a comprehensive and detailed analysis of citizen motivations and engagement mechanisms is reported in Geoghegan et al. (2016), Gharesifard and Wehn (2016) and Rutten et al. (2017) are being studied in detail in the H2020 GroundTruth 2.0 Project

(www.gt20.eu). A possible solution for collecting water level data over time could be the involvement of the civil protection volunteers. This approach is currently being used in the Bacchiglione catchment by the Alto Adriatico Water Authority which requests the water level data at particular location and time to the Civil Protection to validate model results in near real-time. This study demonstrates that high performance model can still be achieved even for decreasing involvement over time. Moreover, crowdsourced observations of either experts or citizens will not necessarily have the quality high enough to support

decision-making (Cortes Arevalo et al., 2014). In addition, real time observations require to ensure safety conditions, internet connection and trusted observers by water authorities. Therefore, it is of utmost importance to understand limitations and to develop quality control mechanisms of CS observations (Tulloch and Szabo, 2012; Vandecasteele and Devillers, 2013; Bordogna et al., 2014; Bird et al., 2014; Cortes Arevalo, 2016).

## 7 Conclusions

This study assess the modelling usefulness of assimilating crowdsourced (CS) observations coming from a network of distributed static physical (StPh), static social (StSc) and dynamic social (DySc) sensors, installed within the WeSenseIt Project in the Bacchiglione catchment, with the aim of advancing in the understanding of the effect of public involvement on the improvements of flood models. In the complex process of assimilating of CS observations in water system models many factors play an important role for the correct flood estimation: type of social sensor, citizen involvement, decrease of the involvement

over time, type of hydrological and hydraulic model, spatial variability of citizen involvement, etc. In this study, we focus on the type of social sensor, on the citizen involvement level and its variability in time and space. The assessment is done for the prediction of the May 2013 flood event in the Bacchiglione catchment, so general conclusions cannot be derived based on one case study only. Because CS observations of water level are not available at the time of this study, we use synthetic observations having intermittent measurement intervals and random accuracy in time and space. Two different sets of experiments are

carried out. In experiment 1, crowdsourced observations from StSc and DySc are assimilated with the hydrological and hydraulic model considering to random levels of citizen involvement. On the other hand, in experiment 2 three hypothetical citizen involvement level scenarios are introduced to provide a more realistic representation of the availability of CS observations for the model. Scenarios are based on the combination of population distribution and three types of citizens' motivations to collect data based on Batson et al. (2002): 1) own personal purposes; 2) shared or community interests and 3)

societal benefits. We further assume that CIL affects only observations intermittency rather than accuracy.

Overall, we demonstrate that assimilation of CS observations provided by citizens improves model performance. Experiment 1.1 shows that assimilation of CS observations in the hydrological model tends to lead to a lower improvement than the

assimilation in the hydraulic model, in case of low lead time values. In case of high lead values, assimilation in the hydrological model allows to achieve better model predictions than the assimilation in the hydraulic model. In Experiment 1.2, high values of $N_{SE}$ are achieved for DySc sensors located close to the boundary conditions, while moving these sensors to downstream locations reduces $N_{SE}$ values. This results are due to the higher error of the boundary conditions if compared to the model error of the hydraulic model itself. Systematic (Bias) and random errors in water level observations plays an important role. Finally, Experiment 2 demonstrates that crowdsourced observations provided by citizens driven by feeling of belonging to a community of peers (motivation 2) can help to improve flood prediction if such small communities are located in the upstream part of the catchment. On the other hand, growing participation of citizens motivated by own purposes, sharing hydrological observations in big cities can help improving model performance. In particular, the model results can benefit from the additional observations provided by enthusiastic citizens. In this study, higher smartphone penetration in the highly urbanized area of Vicenza than in the upstream towns tends to not significantly affect model results. The reduction of citizen involvement over time directly affects the model results. High model performance can still be achieved even for decreasing involvement over time.

A number of limitations of this study have to be addressed as well. Firstly, in order to generalize the findings of this research, the proposed methodology has to be applied in more case studies and flood events. Secondly, real CS observations should be used to properly assess the observational error and accuracy level which vary according to the sensor type (static or dynamic). Thirdly, no specific spatial sensor trajectory of the citizens moving from one StSc sensor to another or using DySc sensors is considered, since this would require the introduction of assumptions about citizens' behaviour during a flood event. This component would be extremely important in the case of dynamic sensors but it could not be included in this research due to the lack of information about citizen involvement in monitoring river water level in the case study. Finally, in real life conditions, it may occur that active citizens might not be available at the right time, i.e. during a flood event. In our study we do not distinguished between observations provided during day time or night time (as addressed by Mazzoleni et al., 2015). For future studies it is recommended to (a) introduce an approach for a better characterization of the CS observations accuracy level, (b) propose an involvement model based on social analysis on citizen motivations and engagement, (c) use agent-based models to simulate and represent the interactions between autonomous agents (citizens) based on their motivations, and (d) test the proposed method using real CS observations during different flood events.

**Appendices**

**Table 7. List of acronyms used in this study**

| Acronyms | Meaning |
|---|---|
| AAWA | Alto Adriatico Water Authority |
| CIL | Citizen Involvement Level |
| CS | Crowdsourced |
| DySc | Dynamic Social |
| KF | Kalman filter |
| MCIL | Maximum Citizen Involvement Level |
| PA | Ponte degli Angeli |
| StPh | Static Physical |
| StSc | Static Social |
| $W_L$ | Water level |
| WSI | WeSenseIt |

**Table 8. Response times for the sub-catchment and the reaches of the Bacchiglione catchment**

| Location | Time (hours) |
|---|---|
| Sub-catchment A | 1.5 |
| Sub-catchment B | 3.5 |
| Sub-catchment C | 6.0 |
| Reach 1 | 2.2 |
| Reach 2 | 2.0 |
| Reach 3 | 7.2 |
| Reach 4 | 9.5 |
| Reach 5 | 3.4 |
| Reach 6 | 5.2 |

**Acknowledgements**

This research was partly funded by the European FP7 Project WeSenseIt: Citizen Observatory of Water, grant agreement No. 308429. Methodological framework development was partly supported by the Russian Science Foundation (grant no. 17-77-30006), and by the IHE Delft Hydroinformatics Research Fund. Data used were supplied by the Alto Adriatico Water Authority.

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
