# Peer review of "Exploring the influence of citizen involvement on the assimilation of crowdsourced observations: a modelling study based on the 2013 flood event in the Bacchiglione catchment (Italy)"

_Hydrology and Earth System Sciences, 2017_

## Referee Comment (RC1) · Anonymous Referee #1 · 1 Mar 2017

GENERAL COMMENT

The paper aims at assessing the usefulness of assimilating crowdsourced observations for improving model-based predictions of flood events, by distinguishing the contribution from static physical sensors from the one derived from either static or dynamic social sensors. Each family of sensors is characterized by a different level of reliability and time of availability. The application of the analysis using hypothetical data to the extreme flood event in the Bacchiglione catchment on May 2013 (when no real crowdsourced observations were available) show a good potential for including this novel

type of information in flood control applications. The manuscript is well written and the topic is definitely interesting. However, I suggest the paper needs a major revision to clarify the specific points discussed below.

SPECIFIC COMMENTS

1) Literature Framework: despite the literature about crowdsourcing information and citizens science is relatively new, I believe that the authors should improve the manuscript's introduction to better frame their work within the existing approaches, which are currently only listed in section 2. In my opinion this is key for clarifying the novelty with respect to previous publications by the same authors, particularly the paper "Can assimilation of crowdsourced streamflow observations in hydrological modelling improve flood prediction?" which also has the same case study application but, more generally, with respect to the entire series Mazzoleni et al. (2015a; 2015b; 2016). In addition, such improved analysis of the literature allows reinforcing the value of the results (obtained with hypothetical data) with respect to the few existing applications run over real crowdsourced observations. Practically, I would suggest re-structuring sections 1 and 2 with the purpose of reviewing the existing approaches and of clarifying how the current paper represents a step-forward with respect to other works.

2) Given the focus on the use of crowdsourced observations, part of the results' discussion (e.g., the analysis on the lead time vs location) is relatively basic and would apply to any type of sensor available along the river. I'm not impressed by the fact that observations far from the outlet sections allow increasing the lead-time. I would hence suggest the authors to consider shortening this discussion in favor of a more extensive analysis of pros and cons of using/relying on crowdsourced data (see point 3).

3) A major limitation of the analysis is the lack of real crowdsourced observations. To overcome this issue, I believe the results would need a more extensive discussion about some key aspects that may strongly impact the results in case real data were available: first of all the level of public engagement is crucial and I would recommend
trying to justify the theoretical formulations with respect to some preliminary findings either from the WeSenseIt project or from similar studies in the literature. I'm quite skeptical about the assumption that the 41% mobile phone penetration can be considered a good proxy for estimating a ratio of active citizens equal to 41%. In addition, I would assume this may vary spatially (even though I don't know whether it could be higher in cities or in the rural areas). Moreover, the distinction of the different behaviors seems also quite theoretical and should be somewhat mapped to the specific case study. Finally, it is not clear how many observations are assumed to be available in each experiment. Given the fast dynamics of flood events, the whole process lasts few hours and indeed the maximum lead-time is one day. This temporal dynamics may however represent a strong constraint for collecting crowdsourced observations, because active citizens might not be there at the right time. I would hence recommend discussing the upper and lower limits in terms of number of observations needed to provide an accurate flood forecast.

4) The last point regards the need of discussing two additional key aspects of crowdsourcing (or more in general citizens science) experiments: how to stimulate citizens engagement and how to keep them engaged in the long term. I understand the authors are assuming a kind of self-motivated behavior differentiated according to the level of engagement. However, in the final discussion, I would suggest the authors to comment about possible techniques for motivating citizens in participating to this data collection experiment and increasing their engagement level (e.g., gamification techniques). In addition, it would be nice discussing also about the potential evolution in time of such engagement as many studies observed decreasing levels of engagement in time. How this would affect the overall flood forecasting system? Assuming it is possible to have a good level of engagement in a critical event, how many citizens are expected to remain active until the next flood? Given the case study analyzed in the paper where floods are not frequent, in my opinion this point is critical as I see a high probability of having a lot of people potentially involved just after a catastrophic flood event who will loose interest in time and may not be active anymore at the next flood event.

MINOR POINTS

- There is a quite intensive use of acronyms. I would suggest - if possible - to reduce it and to add a table of acronyms to help the readers

- Page 3, lines 12-13: soil moisture (from AMSR-E) is repeated duplicated

- Page 3, lines 25-27 / Page 4, lines 14-15: the classification of behaviors from Bonney et al. is duplicated

- Page 9, lines 2-3: why the model does not depend on temperature? how evapotranspiration is estimated?

- Page 23, line19: I assume there is an extra N in "allows to achieve higher N $N_{SE}$"

- Page 25, line 19: sigma(NSE) is never defined. I assume it is the standard deviation across the 100 experiments, but this must be explicitly stated.

---

## Referee Comment (RC2) · Anonymous Referee #2 · 3 Apr 2017

This paper on the potential use of citizen science data for flood forecasting is interesting to the readers of HESS but I have several major and some minor comments and concerns.

Major comments

1) The paper describes the results for multiple experiments, for different river stretches, lead times and stations but the multitude of results are never integrated or discussed. In fact, there is almost no discussion of the results at all. The lack of integration of the different results leaves the reader at loss about what the main take home message

or contribution of this work is. This seriously harms the impact of this study and paper. Often the results for different stream sections or sub-catchments are described in detail and while these specific results (and the differences for the sections or sub-catchments) may be of interest for people working in this basin, it is unclear why the results are different and what was learned from these differences that can be used outside this particular basin. Due to this lack of synthesis and discussion, the paper reads a lot more like a report of a modeling study for an agency (or thesis) than a paper for an international journal. Overall, much more integration and discussion of the results are needed and to clearly state what new thing was learned from this study. The paper has 17 figures, many of which contain multiple subplots and look similar. It is hard for the reader to pin-point what the main or most important "take home" figures are. Is there not a way to merge some of the figures or to summarize the results in a more clever way so that it is clear what figure (and thus what result) the reader should remember from this manuscript? The different figures don't integrate and compare the results from the different experiments and therefore it is hard to compare the model simulation results for the different data types and thus to appreciate the value of the different data types.

2) P4L34: It is unclear how this paper is different from the four other papers by the authors on this topic. It would be good to specifically state here (or elsewhere) what is different between this paper and these previous papers and how this paper builds on the work of (and goes further than) these previous papers.

3) Methods: It is not fully clear what data that could come from citizen science observations was used. On P5L11, both water level and precipitation data are mentioned. From the methods (P5L25) it appears that only the water level data are used and the precipitation data are not used (except the precipitation from the standard measurement stations - not the simulated citizen science data) but then on P18L20-21, P29L4, 11 and P33L1 it is suggested that amateur weather observers will take more measurements. Why would amateur weather observers be particularly interested in water

levels? Weather stations don't regularly measure water levels. Or was the weather data used as well? Also, it is not clear when the water level data was converted to streamflow and when it was just used as water level. On P15L21-25 it is suggested that for this experiment the waterlevels were not converted. Were they only used in the hydraulic model or also in the hydrologic model? Or did that depend on the situation/experiment? If so, then it should be explained much better when water level and when streamflow (converted from the water level observations) was used. If sometimes water level and other times streamflow was used, it should be discussed how this hinders comparison of the results for the different experiments.

4) P11L27: In this study, the modeled streamflow was used to obtain the water level and streamflow data. However, when real citizen science water level data are used, a rating curve is needed for every potential measurement location to obtain information on the streamflow. How would you do that? This is crucial information that is needed when this approach would be used with actual citizen science data (rather than this hypothetical or virtual study). However, almost no guidance is given on how this rating curve information would be obtained for the real citizen science case or how the huge uncertainty in any assumed rating curve will affect the model results. This really needs to be addressed to make the proposed approach useful for real cases with citizen science data. On P15L19 it is suggested that cross sections can be derived from natural cross sections elsewhere but cross sections vary hugely. So this will significantly impact the results.

5) Table 2: How were these values chosen? What are they based on? No references or information is given and therefore it cannot be determined if these values are reasonable at all!! Give references to back up these values or describe how they were chosen and why they are considered reasonable.

6) Table 3: What are these alpha values based on? A reference should be given or the choice of these values should be discussed in detail! P13L7: do these values really suggest that if water levels can be measured from a staff gauge at 1 cm increments

that citizen scientists can estimate the distance between the bank and the water level without a staff gauge with a 2-5 cm accuracy? This latter value seems not reasonable to me at all (since already the surface level of the bank probably differs by a few cm).

7) P14L27: Does this indeed mean that for any given time step there is a 40% chance of getting 1 measurement? Even at night? That does not seem realistic. In the figure CEL values of >80% are used. This is certainly not likely. It would be better to at least also zoom in to the much lower and more realistic CEL values. On P25L20 it is mentioned that the results are highly sensitive to the CEL values. This makes it even more important to show only (or mainly) the results for reasonable CEL values!

8) P18L28-29: This is unclear and not logical. In the case that actual citizen science data are used, you don't know which measurement is most accurate and so you can't use this criteria. You would most likely use both measurements. Why was that not done here?

9) The results (e.g fig 5-6) show that the NSE values are low when the lead time is more than one hour. I miss a discussion on how useful these model simulations are for operational flood management. Is a model prediction with an NSE of 0.4 still useful? It seems unlikely to me that roads can be closed and people evacuated with a lead time that is much less than an hour. Currently there is no discussion about this at all – this really should be included. Also why was NSE used as a criteria and not peak water level or peak flow as well, as in the end that is what is most important in flood management.

10) Overall, the paper is not particularly well written. For many sentences, a more direct or less complicated sentence structure could be used. This would make the paper much easier to read. Some of the information is given twice (e.g. P3l25-28 = p4L14-15), other information is not really necessary (e.g. P2L25-28). Elsewhere lists with other studies are given without any information about them, thus also not the important aspects that are relevant for this study (e.g. P3L28-32). In other places, there

are sentences that may be remnants from moving text around or previous versions that don't fit with the content of the remainder of the paragraph at all (e.g. P4L15-19). I suggest that the authors critically read through the manuscript, include missing information but also remove sentences that do not fit (i.e. break the flow) or don't provide any information that is pertinent to this study.

Other specific comments:

* Title: The title doesn't really tell what the paper is about or what the main findings are. I suggest that you consider changing the title to make it much clearer that this is a hypothetical study that assumes that crowd-sourced data is available (using model results as observations).

* P2L13-15: Add references for each of these attempts.

* P2L21-29: Remove this text. This may be useful in a report but is not really necessary in a scientific publication.

* P3L7: Are 'heat flux sensor' data really that widely available and are they really that useful for flood prediction?

* P3L8: Add references!

* P3L25-29: I don't think that it is necessary to include this information. The paper is already very long.

* P3L29-32: Either take this list of references out or tell what these studies have looked at and how this is relevant for this study.

* P5L8: I thought that this was done by the civil protection. Make it clear that this is not an "average citizen".

* P5L12: Isn't the system already operational?

* P5L17: What are typical response times (and/or travel times of the flood wave) for this

catchment? Without any information on this, it is not possible to interpret the results for the different lead times.

* P5L24: Were these traffic disruptions indeed due to flooded streets (or due to landslides, etc)?

* P6L17: I would not use the word 'sensor' in this context. The text will be much clearer when 'observation' is used as no sensors are used in the DySc. This is particularly the case for wording such as on P26L9, 10, 13, where the number of observations is mentioned and not a particular sensor.

* P13L2: A reference is needed here. I don't think that technicians or hydrologists are necessarily better at estimating depths, volumes or flows than other people. In my experience when multiple hydrologists estimate the depth or flow in a river, their estimates still vary widely.

* Figure 4: Make it clearer in the caption that these are hypothetical curves and not based on previous studies. If not, please include the reference.

* Figures 4-5: Use different line styles so that the figure is also clear when printed in black and white.

* Figure 5a: For which lead time is this result? This is not clear from the caption.

* P21-L1-7: This should be part of the methods (not the results).

* P21: Only the mean simulation results are shown and discussed. The variability in the results should at least be mentioned (or shown with an error band in fig 6).

* P23L1-2: Why? This is an interesting result but not discussed. Just saying that results for A are better than for another catchment may be interesting for people working in this basin but not for the readers of HESS. For them it is much more interesting why these results are so different or what can be learned from these differences. Similar on L21-24 (and many other locations throughout the results) what is interesting about this

result for people outside this basin/what can be learned from this?

* P25L1-4: So here only the water level and not the derived streamflow data are used? But doesn't that make that the comparison between the static and dynamic sensor network results more difficult? This is unclear and needs to be discussed in more detail!

* P28L13-15: Don't overstretch your results. This study shows the model results for different chosen engagement levels but does not provide any information about the actual motivation.

* P29L10: Why is no bias assumed? Isn't it likely that when people estimate the distance between the water level and the stream bank, there is a bias in the resulting water depth information?

* P32L16-18: Add why this was the case.

* Conclusion: This is not a conclusion of the results or a summary of the main take home messages but rather a list of things that were done. That is much less useful than an actual conclusion.

* P33L14-16: Yes this is true but not a part of this study so don't include it in the conclusion.

Minor editorial suggestions:

* P1L18: remove 'for model performance' and insert 'for improving model performance' at the end of the sentence.

* P1L19: insert 'of inclusion of social sensor data'

* P1L29-30: try to rewrite this sentence to make is clearer and easier to understand.

* P2L2: do you mean 'maximum' engagement instead of 'minimum engagement'?

* P2L13: remove 'over'

* P2L17-18: replace 'the benefits' by 'how citizen science data could have benefitted' to make it much clearer that this is a hypothetical situation and actual citizen science data were not available for this event.

* P4L2: Rather than 'minimize low' you could say 'maximize accuracies'

* P4L3-14: This part is about engagement and would fit much better at P5L4 (but this requires a sentence to link it to the previous sentence)

* P4L14-15: Double and not necessary – take out

* P4L16-18: Move to P4L2 where it fits much better.

* P5L29 (and elsewhere): replace 'arrival time' by 'measurement interval'

* Table 1: replace 'lecture' by 'reading'

* P21L4: "random uniform' – this is confusing is it random and variable or uniform?

* P21L15: The caption needs to be improved because it doesn't explain the figure (the figure is not clear for someone who only reads the caption).

* P32L28: Rewrite this sentence- it is unclear

---

## Author Comment (AC1) · 31 May 2017

**RESPONSE TO REVIEWER #1**

**Paper: Towards assimilation of crowdsourced observations for different levels of citizen engagement: the flood event of 2013 in the Bacchiglione catchment**

M. Mazzoleni, V.J. Cortes Arevalo, U. Wehn, L. Alfonso, D. Norbiato, M. Monego, M. Ferri and D.P. Solomatine

**General comment**

*The paper aims at assessing the usefulness of assimilating crowdsourced observations for improving model-based predictions of flood events, by distinguishing the contribution from static physical sensors from the one derived from either static or dynamic social sensors. Each family of sensors is characterized by a different level of reliability and time of availability. The application of the analysis using hypothetical data to the extreme flood event in the Bacchiglione catchment on May 2013 (when no real crowdsourced observations were available) show a good potential for including this novel type of information in flood control applications. The manuscript is well written and the topic is definitely interesting. However, I suggest the paper needs a major revision to clarify the specific points discussed below.*

We appreciate reviewer's compliments and suggestions, which were extremely helpful to improve the manuscript. Additional analyses have been included to assess theoretical scenarios on the effect of smartphone penetration and citizen engagement in urban areas. In addition, some parts have been removed to improve the structure and highlight the most innovative aspects of the manuscript. In the updated version, more focus is given on the citizen engagement effects on DA. The number of scenarios in Experiment 2 are reduced from 6 to 3.

**Specific comments**

**RC 1:** *Literature Framework: despite the literature about crowdsourcing information and citizens science is relatively new, I believe that the authors should improve the manuscript's introduction to better frame their work within the existing approaches, which are currently only listed in section 2. In my opinion this is key for clarifying the novelty with respect to previous publications by the same authors, particularly the paper "Can assimilation of crowdsourced streamflow observations in hydrological modelling improve flood prediction?" which also has the same case study application but, more generally, with respect to the entire series Mazzoleni et al. (2015a; 2015b; 2016). In addition, such improved analysis of the literature allows reinforcing the value of the results (obtained with hypothetical data) with respect to the few existing applications run over real crowdsourced observations. Practically, I would suggest re-structuring sections 1 and 2 with the purpose of reviewing the existing approaches and of clarifying how the current paper represents a step-forward with respect to other works.*

**AC 1:** We thank the reviewer for this critical comment. Indeed, we realized that sections 1 and 2 should be re-structured, and in the revised version of the manuscript we have done that. On the one hand, section 1 (introduction) in the revised version focuses now on the problem definition, the description and limitations of previous studies that include crowdsourced observations in flood modelling and the novelty of the manuscript with respect to the previous one. On the other hand, section 2 now aims at describing theoretical considerations of crowdsourced observations and citizen engagement as basis for

the engagement scenarios. We believe that the manuscript now reads better due to the reviewer's comments.

**RC 2:** *Given the focus on the use of crowdsourced observations, part of the results' discussion (e.g., the analysis on the lead time vs location) is relatively basic and would apply to any type of sensor available along the river. I'm not impressed by the fact that observations far from the outlet sections allow increasing the lead-time. I would hence suggest the authors to consider shortening this discussion in favour of a more extensive analysis of pros and cons of using/relying on crowdsourced data (see point 3).*

**AC 2:** We thank the reviewer for this comment, which have led to several changes in the revised manuscript, as follows. First, we have removed the experiments on assimilation of observations from physical static sensors (experiment 1 in the manuscript) to favour a wider and more detailed analysis on the effect of different engagement levels in the assimilation process (experiments 2 and 3). Second, the paper have been shortened by removing some results in experiments 2 and 3 related to the influence of social sensor location on the assimilation performances. Finally, we have included a discussion section where pros and cons of using crowdsourced observations are considered.

**RC 3:** *A major limitation of the analysis is the lack of real crowdsourced observations. To overcome this issue, I believe the results would need a more extensive discussion about some key aspects that may strongly impact the results in case real data were available: first of all the level of public engagement is crucial and I would recommend trying to justify the theoretical formulations with respect to some preliminary findings either from the WeSenseIt project or from similar studies in the literature. I'm quite skeptical about the assumption that the 41% mobile phone penetration can be considered a good proxy for estimating a ratio of active citizens equal to 41%. In addition, I would assume this may vary spatially (even though I don't know whether it could be higher in cities or in the rural areas). Moreover, the distinction of the different behaviours seems also quite theoretical and should be somewhat mapped to the specific case study. Finally, it is not clear how many observations are assumed to be available in each experiment. Given the fast dynamics of flood events, the whole process lasts few hours and indeed the maximum lead-time is one day. This temporal dynamics may however represent a strong constraint for collecting crowdsourced observations, because active citizens might not be there at the right time. I would hence recommend discussing the upper and lower limits in terms of number of observations needed to provide an accurate flood forecast.*

**AC 3:** We agree with the reviewer that the model used to represent citizen engagement levels is rather theoretical, and we have clearly stated this. In the framework of WeSenseIt project we have carried out an exercise with (52) interested (engaged) citizens who were providing water level observations via the smartphone app to initiate the citizen observatory. However, no formal assessment of citizens' engagement under preparedness and emergency situations has been carried out for this case study. In relation to the Bacchiglione case study, the WeSenseIt Project focused of the analysis of the role of personal weather stations in sharing data via online amateur weather networks, but this data was not neither. Under the consideration that the Civil Protection of Vicenza was involved within the WeSenseIt project, and in reality there is a high chance that it will be happening in the future, however, for this paper it was assumed that only volunteers and/or trained volunteers will participate in providing water level observations. A further assumption is that the mobile application available for the project is easy-to-use and accessible for all participants (this in fact may increase the assumed level of engagement).

The engagement scenarios adopted in our study are assumed to represent the hypothetical situations where citizens will be fully engaged by the Alto Adriatico Water Authority within the Citizen Observatory project. CS observations are assumed to be collected at a particular location and time upon

request from the Alto Adriatico Water Authority. The distinction between favourable attitudes are treated from a theoretical point of view, based on Batson et al. (2002): 1) own personal purposes (usefulness of the collected data for personal interest or direct flood risk management impact); 2) shared or community interests belonging to a community of peers with shared interested; and 3) altruism (beneficing society at large).

In order to clarify this aspect, in the revised manuscript we have included an additional sub-section where citizen engagement level in collecting water level data within the Bacchiglione catchment is described.

Regarding the assumption of 41% for the mobile phone penetration, we have considered this percentage based on Statistica (2016). This value depends on the geographic area and on the characteristic of the population. In fact, we assumed that not everyone will be willing to use smartphone to collect and share water level data due to for example their age and habits. More exhaustive analysis should be performed in order to better define the percentage of active citizens. We agree with the reviewer on the fact that the percentage of active citizen can change spatially. That is why we have conducted additional analysis considering higher percentage of smartphone users (80%) in areas with higher population density, i.e. the municipality of Vicenza. The results of such analysis are reported below (figure 1), and included in the updated version of the manuscript.

[Figure]

*Figure 1. Difference between μ(NSE) values obtained considering standard and higher active citizen percentage in the municipality of Vicenza for different engagement levels from StSc and DySc*

In figure 1, the difference between the $\mu(N_{SE})$ values obtained assuming smartphone penetration of 41% and 80% are represented. From the previous graphs it can be seen that increasing the smartphone penetration in Vicenza does not affect model results in case of scenario 2. In fact, for this scenario, no engagement is assumed in densely urbanized areas such as the municipality of Vicenza. On the other hand, higher number of smartphones in Vicenza affects (partially) only scenarios 1 and 3. In these scenarios we can observe an expected increment in the model performance due to the higher engagement in Vicenza. However, small increments in the NSE values are reported in figure 1, with a maximum difference of 0.04 between normal and higher smartphone penetration. Even in case of the 41% of smartphone usage, high engagement level is achieved for small number of active citizen (see figure 3 of the manuscript). This means that in the proposed theoretical engagement model more active citizens (more mobiles available) won't significantly improve engagement and affect the model performance. In addition, regardless of the number of CS observations if the quality is not good enough the model performance will not improve further improve as described in Mazzoleni et al. (2017). In the revised version, we have included these considerations in the discussions of the results.

The number of observations used in each experiment varies based on a particular engagement level scenario used. In fact, considering a 48h flood event and an hourly model time step, an engagement equal to 1 corresponds to 48 available observations. An engagement of 0.5 implies 24 observations (randomly distributed in time and space) to assimilate. As the reviewer correctly pointed out, active

citizens might not be always available during a flood event. A limitation of our study is that we did not consider the fact that citizen may not provide observations for instance during night hours, as done in an earlier paper by Mazzoleni et al. (2015). This limitation will be mentioned in the concluding section of the updated manuscript.

Finally, it is difficult to recommend a defined number of observations needed to achieve an accurate flood forecast. In fact, as mentioned in Mazzoleni et al. (2017), it is not possible to define a-priori the upper and lower limits of crowdsourced data needed to improve a model because it depends on the uncertainty level of CS observations (which depends on the training level and experience of citizens).

**RC 4:** *The last point regards the need of discussing two additional key aspects of crowdsourcing (or more in general citizen science) experiments: how to stimulate citizens engagement and how to keep them engaged in the long term. I understand the authors are assuming a kind of self-motivated behaviour differentiated according to the level of engagement. However, in the final discussion, I would suggest the authors to comment about possible techniques for motivating citizens in participating to this data collection experiment and increasing their engagement level (e.g., gamification techniques). In addition, it would be nice discussing also about the potential evolution in time of such engagement as many studies observed decreasing levels of engagement in time. How this would affect the overall flood forecasting system? Assuming it is possible to have a good level of engagement in a critical event, how many citizens are expected to remain active until the next flood? Given the case study analysed in the paper where floods are not frequent, in my opinion this point is critical as I see a high probability of having a lot of people potentially involved just after a catastrophic flood event who will lose interest in time and may not be active anymore at the next flood event.*

**AC 4:** We agree with the reviewer's comment on the importance of stimulating citizens' engagement for a long period of time. Many studies reported a temporal pattern of citizen participation driven by self-directed motivations and person's interests. Engagement during flood event tends to disappear if no other event will occur in a short time. In fact, depending on the memory of the community, the awareness of flood risk decays with time, and, therefore, the tendency to be engaged in data collection will also tend to reduce or even disappear. As the reviewer stated correctly, many citizens may be potentially involved just after a catastrophic flood event but they might not be active anymore at the next flood event because they lost interest in time. A possible solution for collecting water level data over time could be the involvement of staff of the civil protection agency which act as (trained) volunteers. This approach is currently being used in the Bacchiglione catchment by the Alto Adriatico Water Authority, which requests the water level data at particular location and time from the Civil Protection to validate model results in near real-time.

As our study is based on theoretical considerations, we simply acknowledged in the discussion the importance to keep citizens engaged. We suggest, for example, gaming approaches or periodic meetings/seminars with interested participants. A more critical and detailed analysis of citizen engagement motivations is reported in Geoghegan et al. (2016).

As the reviewer mentioned, in this study the citizen engagement is assumed constant in time because only one flood event is considered. However, when multiple flood events are simulated, some model has to be used to represent the possible decay of engagement level on time. A possible way to represent such decay is to use a logistic curve and to vary the value of growth rate $r$ over time. Figure 2 (which will be included in the revised version of the manuscript) presents sensitivity analysis of model results with respect to the varying values of the coefficient $r$ to representing the varying engagement levels over time (see in Eq.15). Only engagement scenario 3, for three different values of $w$, is considered. The results demonstrated that decreasing engagement over time (low values of $r$) will lead to a reduction of the model performance, and consequently influence flood prediction. This is somehow an expected

result that, once more, demonstrates the importance of keeping citizens engaged not only for a short period of time but on the long run. However, such reduction of model performances is significant only for values of *r* lower than 0.3, leading to the conclusion that model performances can still be high even if engagement reduces over time up to a given threshold value. Additional relevant literature about engagement strategies are included in the updated version of the manuscript. The ongoing H2020 project GroundTruth2.0 is trying to answer these questions and further research will take those outcomes into account.

[Figure]

*Figure 12. μ(NSE) and σ(NSE) values obtained considering varying values of the coefficient r for scenarios 1 and 3 with three different values of w*

**Minor points**

- *There is a quite intensive use of acronyms. I would suggest - if possible - to reduce it and to add a table of acronyms to help the readers*

As suggested by the reviewer, we have included a table with all the acronyms used in this study.

| | |
|---|---|
| **AAWA** | Alto Adriatico Water Authority |
| **CEL** | Citizen Engagement Level |
| **CS** | Crowdsourced |
| **DySc** | Dynamic Social |
| **KF** | Kalman filter |
| **MCEL** | Maximum Citizen Engagement Level |
| **PA** | Ponte degli Angeli |
| **StPh** | Static Physical |
| **StSc** | Static Social |
| **WL** | Water level |
| **WSI** | WeSenseIt |

- *Page 3, lines 12-13: soil moisture (from AMSR-E) is repeated duplicated*

We have reduced the introductory section and removed the duplicated term

- *Page 3, lines 25-27 / Page 4, lines 14-15: the classification of behaviours from Bonney et al. is duplicated*

We thank the reviewer for spotting this mistake. We have removed the duplicated term

- *Page 9, lines 2-3: why the model does not depend on temperature? how evapotranspiration is estimated?*

In order to shorten the manuscript, we did not provide many details about the hydrological model. For this reason, we referred to Ferri et al (2012) and Mazzoleni et al. (2107) for the interested readers. The temperature is used for the estimation of the real evapotranspiration, which is calculated using the formulation of Hargreaves and Samani (1985)

Reference: Hargreaves, G.H., and Z.A. Samani. 1985. Reference crop evapotranspiration from temperature. Applied Engrg. in Agric. 1:96-99.

- *Page 23, line19: I assume there is an extra N in "allows to achieve higher N N_{SE}"*

Thank you for the comment. We have removed the additional N

- *Page 25, line 19: sigma(NSE) is never defined. I assume it is the standard deviation across the 100 experiments, but this must be explicitly stated.*

We clearly defined sigma(NSE) in the updated version of the manuscript

**Reference:**

Geoghegan, H., Dyke, A., Pateman, R., West, S. & Everett, G. (2016) Understanding motivations for citizen science. Final report on behalf of UKEOF, University of Reading, Stockholm Environment Institute (University of York) and University of the West of England.

Gharesifard, Mohammad, Uta Wehn, and Pieter van der Zaag
 2017    Towards Benchmarking Citizen Observatories: Features and Functioning of Online Amateur Weather Networks. Journal of Environmental Management 193: 381–393.

---

## Author Comment (AC2) · 31 May 2017

**RESPONSE TO REVIEWER #2**

**Paper: Towards assimilation of crowdsourced observations for different levels of citizen engagement: the flood event of 2013 in the Bacchiglione catchment**

M. Mazzoleni, V.J. Cortes Arevalo, U. Wehn, L. Alfonso, D. Norbiato, M. Monego, M. Ferri and D.P. Solomatine

**General comment**

This paper on the potential use of citizen science data for flood forecasting is interesting to the readers of HESS but I have several major and some minor comments and concerns.

We appreciate the critical reviewer's comments and suggestions on our paper. We have addressed them all, in some cases by adding new experiments and in others re-structuring, clarifying and/or removing text and acknowledging the theoretical approach of our study. We therefore believe that the manuscript has improved substantially. Below, you can find a point-to-point discussions of all the comments.

**Major comments**

**RC1:** The paper describes the results for multiple experiments, for different river stretches, lead times and stations but the multitude of results are never integrated or discussed. In fact, there is almost no discussion of the results at all. The lack of integration of the different results leaves the reader at loss about what the main take home message or contribution of this work is. This seriously harms the impact of this study and paper. Often the results for different stream sections or sub-catchments are described in detail and while these specific results (and the differences for the sections or subcatchments) may be of interest for people working in this basin, it is unclear why the results are different and what was learned from these differences that can be used outside this particular basin. Due to this lack of synthesis and discussion, the paper reads a lot more like a report of a modelling study for an agency (or thesis) than a paper for an international journal. Overall, much more integration and discussion of the results are needed and to clearly state what new thing was learned from this study. The paper has 17 figures, many of which contain multiple subplots and look similar. It is hard for the reader to pin-point what the main or most important "take home" figures are. Is there not a way to merge some of the figures or to summarize the results in a more clever way so that it is clear what figure (and thus what result) the reader should remember from this manuscript? The different figures don't integrate and compare the results from the different experiments and therefore it is hard to compare the model simulation results for the different data types and thus to appreciate the value of the different data types.

**AC 1:** We thank the reviewer for the comprehensive comment, and indeed noting the deficiencies of way material is presented in the manuscript. In the complex process of assimilation of crowdsourced observations in water system models many factors play an important role in the correct flood estimation. Those are for example the type of a social sensor, citizen engagement, decay of the engagement over time, type of hydrological and hydraulic model, and quality control of CS observations. We agree with the reviewer and accept that results were presented without giving the reader a clear "take home" message. In this study we focused on the type of social sensors, on the citizen engagement level and its variability in time and space. To better integrate the results and to give a stronger message on the effect of CEL on flood prediction, we removed figures 6, 9, 10, 12, 16 and 17, which contains many multiple sub-plots.

We have included additional analyses on the effect of diminishing citizen engagement in time and variable spatial smartphone penetration on the model performances. In addition, we have reduced the number of scenarios in the experiment 3. In the revised manuscript version we will reduce the number of presented scenarios from 6 to 3 (see next figure). Instead, we will include new analyses on the effect of temporal and spatial variability of citizen engagement levels. Finally, we will divide results and discussions into two separate sections to clearly present the findings of this study.

Figure 1. Original 6 scenarios (left side) and new 3 scenarios (right side) theoretical engagement curves

The main "take home" message we wanted to convey is: assimilation of CS observations provided by citizens improves model performance, and we can show how much, and how this improvement depends on the level of engagement. In particular, assimilation of CS observations in hydrological model tends to lead to a lower improvement than the assimilation in hydraulic models. Bias in water level observations plays an important role. Finally, the effect of spatial variability of active citizens and the decay of citizen engagement in time it is of vital importance to keep adequate model results. This study demonstrates that high model performance can still be achieved even for decreasing engagement in time. We will make these statements clearer in the revised version.

**RC2:** *P4L34:* It is unclear how this paper is different from the four other papers by the authors on this topic. It would be good to specifically state here (or elsewhere) what is different between this paper and these previous papers and how this paper builds on the work of (and goes further than) these previous papers.**

**AC 2:** We thank the reviewer for this valuable comment. Indeed, we should have been clearer. In the revised version, we are clarifying this aspect in the discussion section. In the four previous studies we investigated the effects of assimilating real-time (synthetic) crowdsourced (CS) observations into *hydrological* models. However, in Mazzoleni et al. (2015; 2017a and 2017b), we have not investigated the effect of assimilating CS observations into hydraulic models. Furthermore, we have not considered neither (theoretical) scenarios of citizen engagement, nor the simultaneous assimilation of CS observations from static and dynamic social sensors. For this reason, the main objective of this study is to assess the usefulness of assimilating CS observations in both hydraulic and hydrological models-based predictions of flood events. To that end, we analyse the flood event occurred in May 2013 in the Bacchiglione basin with the data which would come from a distributed network of static physical (StPh),

static social (StSc) and dynamic social (DySc) sensors (however this data was simulated). These (synthetic) CS observations for water level are assimilated in a cascade of hydrological and hydraulic models. The experiments are analysed as CS observations and the assessment of citizen engagement levels are yet not operational nor available in the case study. CEL is further defined as the probability of receiving a CS observation based on the citizen's own interest or intention in collecting water levels. We assume that CEL mainly limit the intermittency or timely availability of observations. All mentioned above will be reflected in the revised manuscript and we hope will improve its clarity.

**RC 3:** Methods: It is not fully clear what data that could come from citizen science observations was used. On P5L11, both water level and precipitation data are mentioned. From the methods (P5L25) it appears that only the water level data are used and the precipitation data are not used (except the precipitation from the standard measurement stations - not the simulated citizen science data) but then on P18L20-21, P29L4, 11 and P33L1 it is suggested that amateur weather observers will take more measurements. Why would amateur weather observers be particularly interested in water levels? Weather stations don't regularly measure water levels. Or was the weather data used as well? Also, it is not clear when the water level data was converted to streamflow and when it was just used as water level. On P15L21-25 it is suggested that for this experiment the water levels were not converted. Were they only used in the hydraulic model or also in the hydrologic model? Or did that depend on the situation/experiment? If so, then it should be explained much better when water level and when streamflow (converted from the water level observations) was used. If sometimes water level and other times streamflow was used, it should be discussed how this hinders comparison of the results for the different experiments.

**AC 3:** We fully agree with this comment, and regret the confusion. On P5L11 we were referring to the mobile app developed within the WeSenseIt project, used to observe both water levels and precipitation. However, in this study we only used water level, since precipitation data are provided by static physical sensors within the catchment. We clarify this aspect in the revised version of the manuscript.

We referred to amateur weather observers to point at the active citizens which will (regularly) provide water level data driven by moral norms and the wish to create knowledge about the hydrological status of the river. We agree with reviewer that the term amateur weather observers might be confusing. For this reason, in the revised version we have modified the term "weather enthusiast individuals" into "enthusiast individuals" which do not use any weather stations for providing water level observations. No weather data was used from these hypothetical citizens, just water level.

In both hydrological and hydraulic models, water level observations are assimilated. Ideally, water level values can be directly assimilated into the hydraulic model. However, in this study we use a Muskingum-Cunge model which requires flow information rather than water level. For this reason, water level value at a certain random location is converted into flow value using Manning equation. Similarly, for the hydrological model the water level observations at the outlet of each sub-catchment has to be converted into streamflow values to be assimilated. This can be done by means of the rating curves available at the sub-catchment outlet cross-sections. As suggested by the reviewer, in the revised manuscript we are providing more details and explanation on the assimilation of water level and streamflow within the hydraulic and hydrological models.

**RC 4:** *P11L27:* In this study, the modelled streamflow was used to obtain the water level and streamflow data. However, when real citizen science water level data are used, a rating curve is needed for every potential measurement location to obtain information on the streamflow. How would you do that? This is crucial information that is needed when this approach would be used with actual citizen science data (rather than this hypothetical or virtual study). However, almost no guidance is given on how this rating

curve information would be obtained for the real citizen science case or how the huge uncertainty in any assumed rating curve will affect the model results. This really needs to be addressed to make the proposed approach useful for real cases with citizen science data. On P15L19 it is suggested that cross sections can be derived from natural cross sections elsewhere but cross sections vary hugely. So this will significantly impact the results.

**AC 4:** As the reviewer correctly mentioned, it is quite unlike to have the information of the rating curve at a random location of the CS observation provided by dynamic sensors in real world applications. In this case, a properly calibrated Manning equation can be used along the river to convert water level into streamflow. Roughness parameter in the Manning equation is calibrated by comparing the observed and simulated rating curve at the outlet section of the catchment. Such a calibrated value is optimal for the cross section of Vicenza but may not be optimal for other upstream sections. In addition, Manning equation uses the cross-section information to estimate hydraulic variables like wetted area and perimeter. When there are no data regarding the cross-section, assumptions should be made about a rectangular cross-section with a given width and depth. Obviously, in case of a more complex hydraulic model, the estimation of streamflow from water level is not required since it can directly assimilate water level.

On the other hand, in case of static sensors the water level can be converted into streamflow using a rating curve. During the installation of the sensor/staff gauge it would be possible to derive the rating curve and cross section in order to convert water level into discharge. However, both Manning equation and rating curve introduce a significant degree of uncertainty in the streamflow estimation. For this reason, CS observations from social sensors are assumed to have lower (and variable) accuracies. These considerations have been included in the conclusion section of the updated manuscript.

**RC 5:** *Table 2: How were these values chosen? What are they based on? No references or information is given and therefore it cannot be determined if these values are reasonable at all!! Give references to back up these values or describe how they were chosen and why they are considered reasonable.*

**AC 5:** In this study we assumed that observations from DySc sensors are randomly biased adding a white noise with standard deviation proportional to a coefficient  $\gamma$ . This coefficient has the same absolute value of the error value of  $\alpha$  in case of StSc sensors. No reference to the choice of  $\gamma$  was provided in the manuscript since it was subjectively assumed. Obviously, we do not want to conclude that 0.3 should be considered as default value to estimate bias in real-life crowdsourced observations. Such bias has to be defined based on field experiments with volunteers proving water level observations during real flood conditions. The main point of this analysis was to provide a sensitivity analysis of model results based on a subjective value of  $\gamma$ .

This study demonstrated that the effect of biased observations on flood prediction strongly depends on the model performance without any assimilation. In the flood event we considered, model without update tends to underestimate the observed water level and streamflow. For this reason, assimilation of overestimated water level data provides higher model performances ( $N_{SE}$ ). On the other hand, underestimation of water level data will give lower NSE. These results can be seen in figure 7 of the paper. These considerations have been included in the conclusion section of the updated manuscript.

**RC 6:** Table 3: What are these alpha values based on? A reference should be given or the choice of these values should be discussed in detail! P13L7: do these values really suggest that if water levels can be measured from a staff gauge at 1 cm increments that citizen scientists can estimate the distance between the bank and the water level without a staff gauge with a 2-5 cm accuracy? This latter value

seems not reasonable to me at all (since already the surface level of the bank probably differs by a few *cm*).

**AC 6:** We agree with the reviewer that the assumption of a citizen scientists estimating the distance between the bank and the water level without a staff gauge with a low error is unrealistic. A more appropriate method for measuring flow at a random location using a dynamic sensor can be the one proposed by Beat et al. (2014). The authors proposed an approach to measure water level with good accuracy, surface velocity and runoff in open channels. However, this approach requires a-priori knowledge on the channel geometry at the random location of the measurement, which is one of the main sources of uncertainty. For this reason, it is assumed that DySc sensors have lower accuracy than StSc. We modified the section 3.1 in the manuscript accordingly.

One of the main and obvious issues in citizen-based observations is to maintain the quality control of the water observations. In this study, the coefficient  $\alpha$  is assumed to be a random variable, uniformly distributed between 0.1 and 0.5 based on the type of the social sensors. The high values of  $\alpha$  for the StSc and DySc sensors are due to the different sources of uncertainty introduced in the water level estimation and the consequent conversion to discharge, since both hydrological and hydraulic models assimilate flow data.

In case of StSc sensors, water level can easily be measured by citizens using a staff gauge as reference. The main source of uncertainty is introduced in the streamflow estimation from water level by means of the Manning equation or available rating curve. The value of  $\alpha$  equal to 0.3 used for StSc is based on our previous studies. In case of DySc sensors, besides the uncertainty in the flow estimation, the assessment of the water level is affected by the uncertainty in the proper knowledge of the cross-section geometry at the random location. For this reason, an error value of 0.5, almost double than for case of StSc, is assumed for DySc sensors.

Unfortunately, we did not have any real crowdsourced observation to test validity of these coefficients. In this study, no sensitivity analysis was performed on the maximum value of  $\alpha$  using dynamic sensors. However, for the revised manuscript, we have already run an additional simulation in which the maximum value of  $\alpha$  is set equal to 0.8 (during Experiment 3) in order to assess the change in model performance due to data assimilation. In this case, the coefficient K on the logistic curve is set equal to 1 (see the  $\mu$ (NSE) values reported in the figure below).

---

## Referee Report (RR1)

Generic:

This manuscript is a useful contribution to the field of hydrology and flood modelling. It also complements similar papers recently published which have used real crowdsourced/citizen science observations to support real hydrological applications. It is evident that the manuscript has been reworked considerably since the original submission and has addressed the reviewers' comments where necessary, which in turn has strengthened the quality of work. However, I still think it is confusing in places, particularly the methods section, and how the synthetic and real-time data were actually generated. I also think that a lot of assumptions are made (which are not always referenced or explained) without fully appreciating the complexity of engaging and involving real citizens. Furthermore, I do not think data quality is fully appreciated; for instance, physical and automatic sensors are subject to error throughout the data collection phase, and not just when level is converted to streamflow. Citizen-based observations are also subject to error in a number of places, particularly if relying on photograph submissions v's a quantitative value. The value of these particular scenarios are therefore limited. It is difficult to apply the findings to a broader scale and be used to influence operational activities. The manuscript reads well in places, but some sentences could do with being rephrased or reworked to improve fluency or clarity. The wider picture needs to be clear and reiterated throughout. The results and accompanying figures are well presented.

Specific (suggested additions in blue italics, text to delete in red italics):

**Page 1, Title:** consider changing 'a model study based in…' to 'a *modelling* study based in...'

**Page 1, Title:** consider either adding '*(Italy)*' to the end of the title so the reader knows where Bacchiglione is, or make it more relevant to a wider audience by removing the place catchment name altogether.

**Page 1, Line 17:** make this clearer. 'less accurate' because non-professionals / the public are collecting valuable hydrological datasets? It needs something to describe what crowdsourcing actually is within the abstract.

**Page 1, Line 19/20:** 'the extreme flood event *which* occurred in'

**Page 1, Line 21:** what do you mean by target point? Do you mean receptor or impact zone? It isn't a very common flood risk management term.

**Page 1, Line 21:** 'Ponte degli Angeli (Vicenza), *at the* outlet of the Bacchiglione catchment'

**Page 1, Line 28/29:** upstream sub-catchment scenarios are very catchment specific. It depends on whether you have a community upstream (less likely) or nearer the outlet (more common). This will affect your results.

**Abstract:** would benefit from documenting more/clearer results in the abstract.

**Page 2, Line 1:** find a better term for 'proper'. What does this even mean?

**Page 2, Line 5:** 'for example, to operate control  structures…'

**Page 2, Line 6:** 'Reliable *and* accurate streamflow simulation…'

**Page 2, Line 7:** 'inherently uncertain due to *the* lack of reliable…'

**Page 2, Lines 7-13:** embed the list of points into the sentence better. E.g. use 'for example..'

**Page 2, Lines 13/14:** 'Data assimilation is a common *method for* updating model input.'

**Page 2, Line 23:** 'citizen-based data *(*Shanley et al…'

**Page 2, Line 29:** 'In both studies, the *observation* filtering process…'

**Introduction:** make it clearer what crowdsourcing actually is and perhaps link them to other similar terms e.g. citizen science ad VGI. Also consider that 'usefulness' doesn't just relate to flood forecasting and real time information. Crowdsourcing can also contribute to, or generate new, long-term datasets over time, and support other types of management activities.

**Page 3, Line 4:** avoid repeating the same word in the sentence ('mentioned')

**Page 3, Line 7:** how is your study real-time? It is not clear. There are reasons why real-time has not been focussed upon e.g. citizens submit their observations at a later date when they have phone signal, wifi or data to submit them.

**Page 3, Lines 4-8:** this text outlines the main research gap for your work (which is good) but are not clearly reflected in your abstract.

**Page 3, Lines 10-13:** 'We analyse a flood event *which* occurred in May 2013 in the Bacchiglione basin *(Italy)* derived from a distributed network of StPh, StSc and DySc sensors. Synthetic CS observations of water level are assimilated in a cascade of hydrological and hydraulic models since real CS measurement are not yet available *for this particular study site*.'

**Page 3, Line 15:** Useful to include a final sentence to say how your papers aim/objectives have a broader relevance.

**Page 3, Line 18:** Useful to include a link to the WeSenseIt project. Don't assume the reader knows what this is.

**Page 3, Lines 23-25**: The project set up a pilot – this is confusing. Makes it sound like citizens were actually involved. If they weren't, who was?

**Page 3, Line 25:** 'usefulness of assimilating CS *WL* observations  to improve the model performance and consequent*ly* flood prediction'.

**Page 3, Line 26:** 'Northern East' should be written as 'North Eastern'. But I would move this to Line 18, when the catchment is first introduced 'The Bacchiglione catchment (*North East* Italy)'.

**Page 3, Line 27/28:** 'river length of about 50km'. Use approximately instead of about.

**Page 3, Line 28:** change left side / right side to east? West?

**Page 4, Line 1:** Forecasted and measured precipitation time series – were these subject to quality assurance and control checks? Are they of a high quality?

**Section 2.1:** Do you need to refer to a location map within this section (i.e. Fig 1)?

**Page 4, Line 6:** 'Three types of sensors *used* to measure WL, static physical (StPh), static social (StSc) and'

**Page 4, Line 9/10:** Any quality control checks for the StPh traditional sensors? Sensors are still subject to error. Why assume? Why not prove this?

**Page 4, Line 13/14:** Is the mobile app used to submit photos, videos and/or quantitative values? Are date, time and location also submitted (i.e. metadata)? Any data quality checks anticipated/required? The app and use of QR codes is very specific and difficult to synthetically generate.

**Page 4, Line 23:** 'We assume a direct relation*ship*…'.

**Page 4, Line 24:** 'i.e. the probability of receiving a CS observation.'

**Page 4, Lines 16-18  / Page 9, Line 13:** estimating velocity and runoff induces significant uncertainty and defeats the object of involving citizens in a cost-effective and simple way. Is it worth the effort if additional data is required or has to be derived? It is unlikely that rating curves would be available in reality. Some studies

are extracting velocities, levels and discharge from videos and photographs automatically using image analysis techniques.

**Page 4, Lines 31/32:** CS activities are not yet operational but this page describes these activities. This makes it confusing to follow. It is not clear how/if synthetic data is used.

**Page 5, Table 1:** How are the photos used? Who extracts the information? Social observations can come in a variety of formats, and is often one of the biggest challenges/barriers when involving citizens. How would this be managed in practice?

**Page 5, Table 1:** Why is StPh regarded as a CS method here? It is automatic and generates the data for you.

**Page 5, Table 1:** Do you have any references to add to the observational error column? Examples do exist in the literature and data quality is important.

**Page 5, Line 9:** 'from a wide range but limited number of' – this is not clear.

**Page 5, Line 10/11:** due to the limited number of participants – isn't that the point? Recruitment and low participation is a huge barrier.

**Page 6, Line 8:** 'In *the* case of the main river channel,'

**Page 6, Figure 1:** it would be useful to mark on the map where the urban area of Vicenza or 'target point' is.

**Page 6, Line 14:** 'Figure 1. Spatial distribution of the sub--catchments, river reaches, *and* StPh and StSc sensors implemented in the catchment by AAWA'

**Page 6, Line 18:** 'relate to the model equation *here* as *a* detailed description is available in Ferri et al. (2012)…'

**Page 6, Line 18/19:** Precipitation time series – can/have the citizens observe this too? Many examples in the literature where they have.

**Page 7, Line 3:** '*T*emperature is used for the estimation'

**Page 7, Line 7:** Information on the quality/success of the calibrated model would be useful. Do you have any statistics to validate its performance?

**Page 10, Line 4:** I do not agree that rating curves are the only source of error/uncertainty. Especially when physical sensors often measure water level indirectly using temperature and pressure.

**Page 10, Line 14:** WL can be easily measured by citizens using a staff – this depends! Some studies have found that their ability to manually observe level using a staff can vary greatly. It can also depend on when it is installed, how turbulent the flow is etc. I feel as though any error associated with the citizens is completely bypassed here. It cannot be assumed that error is the same spatially, temporally and for each participant.

**Page 10, Table 2:** It would be useful to include a citation for the coefficients used in your study, within the table itself or within the table caption so it is clear when they have each come from.

**Page 11, Lines 15-20:** What NSE value do you regard as being 'good' or 'acceptable'?

**Page 13, Line 17:** why have you used 500m and 1000m? Why are they assumed? Citizens may travel or walk elsewhere.

**Page 13, Line 29:** 41% still seems very vague/generic in the context of data submission.

**Page 15, Line 7:** Batson et al 2002 seems an old reference to use for such an evolving topic which is heavily dictated and driven by technology.

**Page 17, Line 8:** Why have you used 80%?

**Page 18, Line 6:** 'and river reaches (hydraulic model) for *a* 1-hour lead time.'

**Page 18, Figure 4:** would be useful to include 'NSE' on or next to the colour ramp key. And repeat for all later figures.

**Page 27, Line 11:** 'so for the assimilation of CS observations it is *also* important to consider  this'

**Page 28, Line 10:** 'This section aims *to summarise*  the main findings of our study and…'

**Page 28, Discussion:** there is scope to relate your findings to the literature in more detail, including those which have used real crowdsourced observations.

**Page 29, Line 23:** 'awareness of flood risk decreases over time' – do you have a reference to back this up?

**Page 29, line 28:** 'Gharesifard and Wehn (2016) *are*  Rutten et al. (2017) and being studied in detail in the H2020 GroundTruth..'

**Page 29, Line 32:** 'This study demonstrates that high *performance models*  can still be achieved even…'

**Page 30, Lines 7-9:** This text is not reflected in the abstract, despite its importance.

**Page 30, Line 13:** Why discuss experiment 2 here and not experiment 1?

**Page 30, Discussion:** what do your results/conclusions mean for the wider picture? Ensure readers can relate to your study.

---

## Author Response (AR2)

**RESPONSE TO REVIEWER #2**

**General comment**

*This manuscript is a useful contribution to the field of hydrology and flood modelling. It also complements similar papers recently published which have used real crowdsourced/citizen science observations to support real hydrological applications. It is evident that the manuscript has been reworked considerably since the original submission and has addressed the reviewers' comments where necessary, which in turn has strengthened the quality of work. However, I still think it is confusing in places, particularly the methods section, and how the synthetic and real-time data were actually generated. I also think that a lot of assumptions are made (which are not always referenced or explained) without fully appreciating the complexity of engaging and involving real citizens. Furthermore, I do not think data quality is fully appreciated; for instance, physical and automatic sensors are subject to error throughout the data collection phase, and not just when level is converted to streamflow. Citizen-based observations are also subject to error in a number of places, particularly if relying on photograph submissions v's a quantitative value. The value of these particular scenarios are therefore limited. It is difficult to apply the findings to a broader scale and be used to influence operational activities. The manuscript reads well in places, but some sentences could do with being rephrased or reworked to improve fluency or clarity. The wider picture needs to be clear and reiterated throughout. The results and accompanying figures are well presented.*

We appreciate the critical reviewer's comments and suggestions to strengthen our contribution. We recognize the need to make explicit the limitations and aspects for further research. We acknowledge that data quality of physical and automatic sensors is subjected to random (e.g. signal noise) and systematic errors (e.g. maintenance needs). In addition, we recognize that visual observations coming from either experts or citizen-based are subject to various errors which may be related to the observer and data collection method (expertise and photography) but also to the specific conditions at the reporting location (accessibility and visibility). We have added a section referring to the limitations of the scenarios analysed which includes recommendation for further research. We recognize that our approach cannot yet be used in operational activities. However, repeated data collection exercises will help to refine these scenarios. Description of the method used to generate synthetic data is improved. In addition, we have also restructured section 2.2 to clearly separate the general description of the sensors implemented used in the WSI project and the specific method and assumption on data accuracy used for this research. We believe that the manuscript has significantly improved and benefit from reviewer suggestions.

The updated text in the manuscript is marked using green colour. Below, you can find a point-to-point discussions of all the comments.

**Specific comments (suggested additions in blue italics, text to delete in red italics):**

**Page 1, Title:** *consider changing 'a model study based in…' to 'a modelling study based in…'*

**Page 1, Title:** *consider either adding '(Italy)' to the end of the title so the reader knows where Bacchiglione is, or make it more relevant to a wider audience by removing the place catchment name altogether.*

Based on reviewer comments we have change the title of the manuscript accordingly.

*Page 1, Line 17: make this clearer. 'less accurate' because non-professionals / the public are collecting valuable hydrological datasets? It needs something to describe what crowdsourcing actually is within the abstract.*

Crowdsourced water levels are recently being considered as complementary data to traditional sensors. These water levels are provided by interested (engaged) citizens with an estimated value and/or photography via smartphone application. These observations are less accurate due to the variable expertise of the observer, qualitative measurement and variable conditions at the reporting location.

*Page 1, Line 19/20: 'the extreme flood event which occurred in'*

We agreed with the suggestions and have accordingly implemented them in the text.

*Page 1, Line 21: what do you mean by target point? Do you mean receptor or impact zone? It isn't a very common flood risk management term.*

Thanks to the reviewer' comment we realized it was not clear the term "target point". That is why we have replaced it with "prediction point" throughout the manuscript.

*Page 1, Line 21: 'Ponte degli Angeli (Vicenza), at the outlet of the Bacchiglione catchment'*

The target point where the water levels are predicted is the outlet of the Bacchiglione. To make it explicit we now simply refer it as: By means of a semi-distributed hydrological model, flood water levels are predicted at the point of Ponte degli Angeli (Vicenza) which is the outlet of the Bacchiglione.

*Page 1, Line 28/29: upstream sub-catchment scenarios are very catchment specific. It depends on whether you have a community upstream (less likely) or nearer the outlet (more common). This will affect your results.*

We agree with the reviewer. Our study demonstrate the usefulness of integrating crowdsourced observations however the results are catchment specific and rely on the quality control procedures of crowdsourced observations. Therefore, we have accordingly mentioned it in the abstract. To specify that these results are case specific we have modified the text of the manuscript as follows:

The results demonstrate the usefulness of integrating crowdsourced observations. First, the assimilation of crowdsourced observations located at upstream points of the Bacchiglione catchment ensure high model performance for high lead time values, whereas observations at the outlet of the catchments provide good results for short lead times. Second, low quality crowdsourced observations can significantly affect model results. Third, for the theoretical

scenario of citizens motivated for their feeling of belonging to a "community of friends", flood prediction is improved when such small communities are located in the upstream portion of the Bacchiglione catchment. Growing participation of citizens motivated by personal interests, sharing hydrological observations can help improving model performance, and therefore effective communication between water authorities and citizens is encouraged. Finally, decreasing involvement over time lead to a reduction of the model performance and consequently inaccurate flood forecasts

**Abstract:** *would benefit from documenting more/clearer results in the abstract.*

Based on reviewer' recommendation, a few lines have been reformulated and added to clarify the results, matching the conclusions. To improve the clarity of the abstract, now results are numbered and aspects for further research are also stated.

**Page 2, Line 1:** *find a better term for 'proper'. What does this even mean?*

A new sentence, more related to the conclusions, has been added. The previous one was not clear and it was removed.

**Page 2, Line 5:** *'for example, to operate control river structures...'*

**Page 2, Line 6:** *'Reliable and accurate streamflow simulation...'*

**Page 2, Line 7:** *'inherently uncertain due to the: lack of reliable...'*

We agreed with the suggestions and have accordingly implemented them in the text.

**Page 2, Lines 7-13:** *embed the list of points into the sentence better. E.g. use 'for example..'*

We thank the reviewer for this suggestion. The introduction has been restructured, among others to address this comment.

**Page 2, Lines 13/14:** *'Data assimilation is a common method for updating model input.'*

**Page 2, Line 23:** *'citizen-based data (Shanley et al...'*

**Page 2, Line 29:** *'In both studies, the observation filtering process...'*

The previous three suggestions have been addressed in the update version of the manuscript

**Introduction:** *make it clearer what crowdsourcing actually is and perhaps link them to other similar terms e.g. citizen science ad VGI. Also consider that 'usefulness' doesn't just relate to flood forecasting and real time information. Crowdsourcing can also contribute to, or generate new, long-term datasets over time, and support other types of management activities.*

We appreciate reviewers' valuable suggestion to relate our study with the diverse terms used in the scientific literature. We have made the necessary additions in the introduction

and have further clarified the way we propose to use crowdsourced observations in our study. A new paragraph with the use of similar terms has been added, and some text has been moved. See below the text that has been added.

"In parallel, the availability of recent technological advances to the public has motivated the idea of involving people in data collection. Various terms have been used for this concept in different areas (Wehn and Evers, 2015). In Natural Science this idea is known as 'citizen science' (e.g, Silvertown, 2009); in Geography, 'volunteer geographic information, VGI' (Goodchild, 2007) and 'crowdsourcing geospatial data' (Heipke, 2010), and in Computer Science 'people-centric sensing' (Campbell et al., 2006) and 'participatory sensing' (Höller et al., 2014). Other terms explicitly emphasise the involvement of the public, for instance the 'value of information and public participation' (Alfonso, 2010), 'public computing' (Anderson, 2003) and 'community data collection' (Aanensen et al., 2000)."

*Page 3, Line 4: avoid repeating the same word in the sentence ('mentioned')*

We have addressed this comment in the manuscript.

*Page 3, Line 7: how is your study real-time? It is not clear. There are reasons why real-time has not been focussed upon e.g. citizens submit their observations at a later date when they have phone signal, wifi or data to submit them.*

Our study is real time as a synthetic dataset of CS observations is assimilated at the time periods in which the flood event in the Bacchiglione occurred. However, we agreed with the reviewer that the assimilation of real time observations bring forth challenges that go beyond the availability of observers at a given location. Real time observations require to ensure safety conditions, internet connection and trusted observers by water authorities. We further elaborate these limitations in the discussion. In the introduction, we refer to our study simply assessing to the usefulness of CS observations while taking into account its variable distribution, intermittency and potentially lower quality at the time period of the flood event.

*Page 3, Lines 4-8: this text outlines the main research gap for your work (which is good) but are not clearly reflected in your abstract.*

We have rephrased the abstract to more explicitly mention these aspects. A new sentence has been added to the abstract:

Unfortunately, just few studies have investigate the benefit of assimilating crowdsourced data in hydrological and hydraulic model.

*Page 3, Lines 10-13: 'To that end, wWe analyse athe flood event which occurred in May 2013 in the Bacchiglione basin (Italy) derived from a distributed network of StPh, StSc and DySc sensors. Synthetic CS observations of water level are assimilated in a cascade of hydrological*

*and hydraulic models since real CS measurement are not yet available for this particular study site.'*

We have incorporated suggestions accordingly and further introduced the acronyms as they are being referred for first time in the introduction.

***Page 3, Line 15:*** *Useful to include a final sentence to say how your papers aim/objectives have a broader relevance.*

We thank the reviewer for this good idea. Introduction concludes now with a sentence that makes broader our contribution while acknowledging the aspects for further research that are also relevant for other fields:

"The achievement of the paper's objective is a step forward in understanding the effect of public involvement on the possible improvement of physical models, with methods that can be replicated in other fields"

***Page 3, Line 18:*** *Useful to include a link to the WeSenseIt project. Don't assume the reader knows what this is.*

We have include the link to the project as suggested

***Page 3, Lines 23-25***: *The project set up a pilot – this is confusing. Makes it sound like citizens were actually involved. If they weren't, who was?*

As suggested, we have rearranged the description of the pilot setup for clarity. We further refer to the section 2.3 that describes the progress and limitations of the citizen involvement in the Bacchiglione catchment.

***Page 3, Line 25:*** *'usefulness of assimilating CS WL observations or WL to improve the model performance and consequently flood prediction'.*

***Page 3, Line 26:*** *'Northern East' should be written as 'North Eastern'. But I would move this to Line 18, when the catchment is first introduced 'The Bacchiglione catchment (North East Italy)'.*

***Page 3, Line 27/28:*** *'river length of about 50km'. Use approximately instead of about.*

***Page 3, Line 28:*** *change left side / right side to east? West?*

The four previous suggestions have been included in the updated version of the manuscript

***Page 4, Line 1:*** *Forecasted and measured precipitation time series – were these subject to quality assurance and control checks? Are they of a high quality?*

We thank the reviewer for pointing out this misunderstanding. Forecasted and measured precipitation time series are available for a flood event that occurred in May 2013. In

particular, the forecasted precipitation time series is provided by the Cosmo-LAMI model, a regional model that provides numerical prediction over the national territory at 7 km resolution and three-day time interval. Currently, AAWA is performing quality control on the forecasted data before using them in the Bacchiglione flood early warning system. On the other hand, the measured precipitations are supplied and validated by Veneto Regional Agency of Environmental Prevention and Protection (ARPAV). We have included this text in the updated manuscript.

***Section 2.1:*** *Do you need to refer to a location map within this section (i.e. Fig 1)?*

We have include the reference to Figure 1.

***Page 4, Line 6:*** *'Three types of sensors used to measure WL, static physical (StPh), static social (StSc) and'*

*The text has been modified.*

***Page 4, Line 9/10:*** *Any quality control checks for the StPh traditional sensors? Sensors are still subject to error. Why assume? Why not prove this?*

The measured river water levels at Ponte Angeli are provided and validated by ARPAV. The levels are measured with an ultrasonic transducer that compensates the echo received basing on the air temperature measured by the integrated thermometer. Certainly the measured levels, even if validated, are associated with an instrumental error. Typically the instrument precision is about 0.01 m.

On the other hand, the conversion from rating curves is subject to significant uncertainties due to the poor reliability of the steady state hypothesis, for the variability of the roughness of the river bed, for periodic variations of the geometry of the river cross section and for extrapolation-induced errors (may be of the order of 20%).

We have reported the quality control check and instrumental precision in the manuscript. Considerations regarding high uncertainty in the rating curve estimation have also been included.

***Page 4, Line 13/14:*** *Is the mobile app used to submit photos, videos and/or quantitative values? Are date, time and location also submitted (i.e. metadata)? Any data quality checks anticipated/required? The app and use of QR codes is very specific and difficult to synthetically generate.*

Concerning the QR code function, the mobile app does not allow to send photos and videos but only the quantitative value of the river level observed at a specific staff gauge. This value is submitted in association with date/time. The location is incorporated in the QR code. The WSI mobile app is equipped with a filter that automatically discards the river level measurements that fall outside the range associate to the staff gauge. We have also restructured section 2.2 to clearly separate the general description of the sensors

implemented used in the WSI project and the specific method and assumption on data accuracy used for this research.

***Page 4, Line 23:*** *'We assume a direct relationship…'.*

***Page 4, Line 24:*** *'i.e. the probability of receiving a CS observations.'*

We have included these changes in the text

***Page 4, Lines 16-18 / Page 9, Line 13:*** *estimating velocity and runoff induces significant uncertainty and defeats the object of involving citizens in a cost-effective and simple way. Is it worth the effort if additional data is required or has to be derived? It is unlikely that rating curves would be available in reality. Some studies are extracting velocities, levels and discharge from videos and photographs automatically using image analysis techniques.*

We agreed with the reviewer on the increasing uncertainty of estimating velocity and runoff with dynamic sensors. In addition, it is extremely difficult to have rating curve information at any random location of the DySc sensors. That is why, a Manning equation for rectangular channel and given channel roughness can be used to derive river flow. However, this approach will introduce significant uncertainty. A possible solution is the use of mobile apps able to retrieve flow observations as described by Luthi et al. (2014). We believe that this type of mobile app will increasingly become available (at reasonable low costs) to citizen in order to easily measure river flow. We have included these considerations in the updated version of the manuscript. We extended the description of Michelsen et al (2016) as example from the literature to overcome these limitations.

***Page 4, Lines 31/32:*** *CS activities are not yet operational but this page describes these activities. This makes it confusing to follow. It is not clear how/if synthetic data is used.*

We agree with the reviewer that this part may look confusing. Currently, only StPh sensors are used by AAWA to provide daily flood forecast in the Bacchiglione catchment. Despites that CS observations were not operational nor available in the case study for the flood event of 2013, we analysed the (potential) characteristics of each sensor to generate the synthetic data used in this study. We have added this clarification at the beginning of section 2.2. We have also removed the reference to the synthetic observations.

***Page 5, Table 1:*** *How are the photos used? Who extracts the information? Social observations can come in a variety of formats, and is often one of the biggest challenges/barriers when involving citizens. How would this be managed in practice?*

Social observations can come in a variety of formats, and it is often one of the biggest challenges when involving citizens. Photos can be automatically analysed using image recognition methods as proposed by van Overloop and Vierstra (2015) and Le Boursicaud et al. (2015), in which a reference gauge must be available. For the WSI project, Quick

Response codes (QR) have been added to gauges with automatic water level sensors, which are read in combination with a dedicated mobile application. No pattern recognition for water level photos is implemented in the WSI mobile app. We have included these considerations in the updated version of the manuscript.

***Page 5, Table 1:*** *Why is StPh regarded as a CS method here? It is automatic and generates the data for you.*

Table 1 refers to all type of observations and not only CS. We agreed with the reviewer and have explicitly described StPh as one of the sensor types, which is not a CS type due to its automatic method for data collection. To clarify this point we have also modified the table caption.

***Page 5, Table 1:*** *Do you have any references to add to the observational error column? Examples do exist in the literature and data quality is important.*

We appreciated reviewer's comment and have accordingly add a column with supporting examples from the literature.

***Page 5, Line 9:*** *'from a wide range but limited number of' – this is not clear.*

We have improved this sentence removing "a wide range but"

***Page 5, Line 10/11:*** *due to the limited number of participants – isn't that the point? Recruitment and low participation is a huge barrier.*

The exercise that was carried out within the framework of the WeSenseIt project had the purpose of testing the infrastructure and set up for CS observations. Despites the limited number of participants, the duration and set up of the exercise did not intend to involve other citizens than the volunteers of Civil Protection. We further clarify that in this study we do not refer to the engagement process (how to get citizens involved) but rather to the probability of receiving a CS observation based on the citizen's own interest or intention in collecting water levels. We agreed that engagement and involvement level are related and represent a huge barrier to collect CS observations therefore we have accordingly referred it in the discussion.

***Page 6, Line 8:*** *'In the case of the main river channel,'*

The sentence has been improved

***Page 6, Figure 1:*** *it would be useful to mark on the map where the urban area of Vicenza or 'target point' is.*

The prediction location of Ponte degli Angeli (PA) corresponds to the StPh-3 sensor. We have included this information in the caption of figure 1.

**Page 6, Line 14:** *'Figure 1. Spatial distribution of the sub--catchments, river reaches, and StPh and StSc sensors implemented in the catchment by AAWA'*

**Page 6, Line 18:** *'relate to the model equation here as a detailed description is available in Ferri et al. (2012)...'*

The text has been modified including the previous suggestions

**Page 6, Line 18/19:** *Precipitation time series – can/have the citizens observe this too? Many examples in the literature where they have.*

Previous studies (de Vos et al., 2017; Starkey et al., 2017) have demonstrated useful CS reports from 24-hours rainfall measurements. However, these measurements are not within the scope of this study as the input precipitation for the hydrological model are hourly instead of 24-hours time series. In the current version of the WSI mobile app citizen may report observed precipitation values. However, this is out of the scope of our paper (as specified in section 2).

**Page 7, Line 3:** *'The tTemperature is used for the estimation'*

Text has been corrected.

**Page 7, Line 7:** *Information on the quality/success of the calibrated model would be useful. Do you have any statistics to validate its performance?*

The hydrological model was calibrated and validated by AAWA. Calibration results are briefly reported in Ferri et al. (2012). We have included this reference in section 3.1.1.

**Page 10, Line 4:** *I do not agree that rating curves are the only source of error/uncertainty. Especially when physical sensors often measure water level indirectly using temperature and pressure.*

We agreed with the reviewer that other sources of uncertainty exist in physical sensors (i.e. ultrasonic) such as temperature fluctuations and turbulence (Irrigation Training and Research Center, 1998, p. 58). However, for simplification purposes we assume that the main source of uncertainty is the rating curve. That is why we have changed "only" by "main".

**Page 10, Line 14:** *WL can be easily measured by citizens using a staff – this depends! Some studies have found that their ability to manually observe level using a staff can vary greatly. It*

*can also depend on when it is installed, how turbulent the flow is etc. I feel as though any error associated with the citizens is completely bypassed here. It cannot be assumed that error is the same spatially, temporally and for each participant.*

We agreed with the reviewer on that random error and bias or systematic errors (Bird et al., 2014) for WL observations vary temporally, spatially and for each type of sensor (physical or social). Moreover, we acknowledge that the distribution of accuracy levels for CS observations is a simplified and first approximation that is aspect for further research

***Page 10, Table 2:*** *It would be useful to include a citation for the coefficients used in your study, within the table itself or within the table caption so it is clear when they have each come from.*

We agreed with the reviewer that the assumptions behind the distribution of accuracy levels should be clarified and it is now listed in bullet points above the Table 2.

***Page 11, Lines 15-20:*** *What NSE value do you regard as being 'good' or 'acceptable'?*

Based on Moriasi et al. (2007), NSE values between 0.0 and 1.0 are generally considered as acceptable levels of model performance.

***Page 13, Line 17:*** *why have you used 500m and 1000m? Why are they assumed? Citizens may travel or walk elsewhere.*

In this study, a spatial discretization of 1000m is used in order to guarantee the numerical stability of the MC model scheme (as added at the end of section 3.1). On the other hand, the value of 500m is selected based on a subjective judgment, assuming that citizens located further than 500m from the river are not contributing to the collection of CS observations. Obviously, Different extents of the buffer will lead to different coverages of the active area, with significant effects on the simulated number of involved citizens. In order to consider more complex citizen behaviours we suggested (in the conclusion section) the use of Agent Based Model as next research step.

***Page 13, Line 29:*** *41% still seems very vague/generic in the context of data submission.*

We recognise this limitation. According to Statistica (2016), the mobile phone penetration in Italy in 2013, the year of the flood event analysed in this study, was about 41%, which means that about 41% of the population was potentially able to submit data. In view of the lack of a better source, we assume that this proportion is valid also for the regional scope. The text in the manuscript has been modified in order to explicitly include this limitation.

***Page 15, Line 7:*** *Batson et al 2002 seems an old reference to use for such an evolving topic which is heavily dictated and driven by technology.*

These citizen involvement scenarios are based on Batson et al., (2002), whose aggregated categories of citizen's motivations are still in agreement with more comprehensive and detailed analysis such the ones recently reported in Geoghegan et al. (2016) and (Gharesifard & Wehn, 2016).

***Page 17, Line 8:*** *Why have you used 80%?*

We have used 80% as a double percentage of active citizens is assumed in this specific analysis. Thanks to reviewer' comment we have clarified this aspect in the text

***Page 18, Line 6:*** *'and river reaches (hydraulic model) for a 1-hour lead time.'*

Text has been corrected accordingly

***Page 18, Figure 4:*** *would be useful to include 'NSE' on or next to the colour ramp key. And repeat for all later figures.*

As requested by the reviewer, we have included NSE on the colour bar of figures 4, 5, 7, 8, 10 and 11.

***Page 27, Line 11:*** *'so for the assimilation of CS observations it is also important to consider also this'*

***Page 28, Line 10:*** *'This section aims to summarise at summarizing the main findings of our study and...'*

We have corrected the text based on reviewer suggestions.

***Page 28, Discussion:*** *there is scope to relate your findings to the literature in more detail, including those which have used real crowdsourced observations.*

We agree with the reviewer. As referred in the introduction, there have been different studies in which crowdsourced observations were used to improve model prediction. In the discussion section we underlined how the findings of our study are in accordance with recent research carried out using real crowdsourced observations. We believe that the comment of the reviewer have strengthen the discussion section and improve the quality of the manuscript.

***Page 29, Line 23:*** *'awareness of flood risk decreases over time' – do you have a reference to back this up?*

We have included Raaijmakers et al. (2008) as reference in the paper:

Raaijmakers, R., J. Krywkow and A. van der Veen (2008). "Flood risk perceptions and spatial multi-criteria analysis: an exploratory research for hazard mitigation." Natural Hazards 46(3): 307-322

***Page 29, line 28:*** *'Gharesifard and Wehn (2016)* are and *Rutten et al. (2017) and being studied in detail in the H2020 GroundTruth..'*

***Page 29, Line 32:*** *'This study demonstrates that high* performance models value of model performance *can still be achieved even…'*

The text of the manuscript has been corrected accordingly

***Page 30, Lines 7-9:*** *This text is not reflected in the abstract, despite its importance.*

We thank the reviewer for this suggestion. We have restructured abstract in order to account for these aspects.

***Page 30, Line 13:*** *Why discuss experiment 2 here and not experiment 1?*

Thanks to the reviewer' comment we have included an additional sentence summarizing the scope of experiment 1 with respect to experiment 2

***Page 30, Discussion:*** *what do your results/conclusions mean for the wider picture? Ensure readers can relate to your study.*

The reviewer is right. The sentence at the beginning of the conclusions has been expanded to include a reflection for the wider picture. We reckon it was needed.